# Tensor Decomposition Networks for Fast Machine Learning Interatomic Potential Computations

**Yuchao Lin[1,2]\***, **Cong Fu[1]\***, **Zachary Krueger[1]**, **Haiyang Yu[1]**, **Maho Nakata[3]**,
**Jianwen Xie[2]**, **Emine Kucukbenli[4]**, **Xiaofeng Qian[5]**, **Shuiwang Ji[1,5,6]†**

[1]Department of Computer Science and Engineering, Texas A&M University, USA
[2]Lambda, Inc., USA
[3]RIKEN Cluster for Pioneering Research, RIKEN, Japan
[4]NVIDIA, USA
[5]Department of Materials Science and Engineering, Texas A&M University, USA
[6]Mike Walker '66 Department of Mechanical Engineering, Texas A&M University, USA

## Abstract

$SO(3)$-equivariant networks are the dominant models for machine learning interatomic potentials (MLIPs). The key operation of such networks is the Clebsch-Gordan (CG) tensor product, which is computationally expensive. To accelerate the computation, we develop tensor decomposition networks (TDNs) as a class of approximately equivariant networks in which CG tensor products are replaced by low-rank tensor decompositions, such as the CANDECOMP/PARAFAC (CP) decomposition. With the CP decomposition, we prove (i) a uniform bound on the induced error of $SO(3)$-equivariance, and (ii) the universality of approximating any equivariant bilinear map. To further reduce the number of parameters, we propose path-weight sharing that ties all multiplicity-space weights across the $\mathcal{O}(L^3)$ CG paths into a single shared parameter set without compromising equivariance, where $L$ is the maximum angular degree. The resulting layer acts as a plug-and-play replacement for tensor products in existing networks, and the computational complexity of tensor products is reduced from $\mathcal{O}(L^6)$ to $\mathcal{O}(L^4)$. We evaluate TDNs on PubChemQCR, a newly curated molecular relaxation dataset containing 105 million DFT-calculated snapshots. We also use existing datasets, including OC20, and OC22. Results show that TDNs achieve competitive performance with dramatic speedup in computations. Our code is publicly available as part of the AIRS library (https://github.com/divelab/AIRS/).

## 1 Introduction

Symmetry is a fundamental aspect of molecular and material systems [1], making it a crucial consideration in developing machine learning interatomic potentials (MLIPs). Equivariant graph neural networks have emerged as dominant frameworks in this domain. Usually, equivariant models incorporate directional features and spherical harmonics to maintain equivariance under rotation symmetry [2, 3, 4]. Among these, tensor product (TP) operations play a central role in fusing equivariant features, providing a powerful mechanism for building expressive models that adhere to $SO(3)$-equivariance [5, 6, 7, 8, 9].

However, the computational cost of tensor product operations grows rapidly with the maximum angular degree $L$, reaching $\mathcal{O}(L^6)$ in conventional implementations. Recent efforts to mitigate this

---

*Equal contribution
†Correspondence to: Shuiwang Ji <sji@tamu.edu>

cost have focused on accelerating the TP operation or applying frame averaging (FA) to enforce equivariance [10, 11, 12, 13]. While frame averaging is architecture-agnostic, it suffers from discontinuity issues [11]. On the other hand, TP acceleration techniques, such as SO(2)-based convolutions [14, 15] and fast spherical Fourier transformations [16], reduce complexity but are no longer the standard CG tensor product with the same expressivity. Thus, there remains a gap in developing a method that simultaneously reduces computational complexity and parameter count while maintaining similar accuracy and equivariance of CG tensor product.

In this work, we propose Tensor Decomposition Networks (TDNs), a new class of approximately equivariant networks that replace the standard CG tensor product with low-rank tensor decompositions based on CANDECOMP/PARAFAC (CP) decomposition. TDNs introduce a CP decomposition that provides error bounds on equivariance and universality, ensuring consistency under SO(3) transformations. Additionally, a path-weight sharing mechanism consolidates multiplicity-space weights across CG paths, significantly reducing the parameter count from $\mathcal{O}(cL^3)$ to $\mathcal{O}(c)$ with $c$ the parameter count of weight of each path. The resulting layer has expressive power close to conventional TP while lowering computational complexity from $\mathcal{O}(L^6)$ to $\mathcal{O}(L^4)$. We validate TDNs on a newly curated relaxation dataset with 105 million DFT-calculated molecular snapshots, along with the established OC20 and OC22 datasets, demonstrating competitive accuracy with substantial computational speedup.

## 2 Preliminaries and Related Work

Clebsch–Gordan (CG) tensor product is a fundamental operation widely used as the backbone for SO(3)-equivariant neural network, enabling the fusion of feature fields at different angular degrees. We discuss the formal definition of CG tensor product with the maximum angular degree $L$. Consider two feature fields $\boldsymbol{x} = \bigoplus_{\ell_1=0}^{L} \boldsymbol{x}^{(\ell_1)}$ and $\boldsymbol{y} = \bigoplus_{\ell_2=0}^{L} \boldsymbol{y}^{(\ell_2)}$, where $\boldsymbol{x}^{(\ell_1)}, \boldsymbol{y}^{(\ell_2)}$ are type-$\ell_1$ and type-$\ell_2$ irreducible representations (irreps) of SO(3). The CG tensor product fuses a pair of SO(3)-representations $\boldsymbol{x}^{(\ell_1)}$ and $\boldsymbol{y}^{(\ell_2)}$ into every admissible output type $|\ell_1 - \ell_2| \leq \ell_3 \leq \ell_1 + \ell_2$ via the CG coefficients $C_{\ell_1,m_1,\ell_2,m_2}^{\ell_3,m_3}$, defined as:

$$(\boldsymbol{x}^{(\ell_1)} \otimes_{CG} \boldsymbol{y}^{(\ell_2)})_{m_3}^{(\ell_3)} = \sum_{m_1=-\ell_1}^{\ell_1} \sum_{m_2=-\ell_2}^{\ell_2} C_{\ell_1,m_1,\ell_2,m_2}^{\ell_3,m_3} \, \boldsymbol{x}_{m_1}^{(\ell_1)} \, \boldsymbol{y}_{m_2}^{(\ell_2)}, \qquad -\ell_3 \leq m_3 \leq \ell_3. \quad (1)$$

A triple $(\ell_1, \ell_2, \ell_3)$ satisfying the CG selection rule $|\ell_1 - \ell_2| \leq \ell_3 \leq \ell_1 + \ell_2$ is referred to as a *path*, and each path constitutes an independent SO(3)-equivariant mapping. Collecting the contributions from all admissible paths, the complete CG tensor product is expressed as:

$$\boldsymbol{x} \otimes_{CG} \boldsymbol{y} = \bigoplus_{\ell_1=0}^{L} \bigoplus_{\ell_2=0}^{L} \bigoplus_{\ell_3=|\ell_1-\ell_2|}^{\min(\ell_1+\ell_2, L)} (\boldsymbol{x}^{(\ell_1)} \otimes_{CG} \boldsymbol{y}^{(\ell_2)})^{(\ell_3)}.$$

However, the computational complexity of the CG tensor product scales as $\mathcal{O}(L^6)$ as it involves $\mathcal{O}(L^3)$ distinct paths and $\mathcal{O}(L^3)$ operations per path. This significant cost poses a major bottleneck in practical implementations, especially when dealing with higher angular degrees. To mitigate this computational challenge, inspired by tensor decomposition [17] to low-rank tensors, we propose an efficient approximation based on tensor decomposition techniques, specifically the CP decomposition, to reduce the complexity while preserving approximate equivariance. The detailed formulation and implementation of the CP decomposition are presented in the following section.

**Invariant and Equivariant Models.** Symmetry has been a widely discussed constraint when developing machine learning methods for predicting chemical properties of molecules. Invariant and equivariant graph models have been widely used in these cases. Invariant models [18, 2, 19, 20, 21] aim to consider the rotation invariant features such as pairwise distance as the input, and make use of these to predict final properties. Equivariant models [3, 4, 22, 23, 24, 25] further incorporate equivariant features such as pairwise directions and spherical harmonics into the model. These models are built with equivariant blocks to ensure that output features rotate consistently with any rotation applied to the input features, thereby maintaining equivariant symmetry.

**Tensor Product Acceleration.** Among these equivariant networks, tensor product [5, 6, 26, 7, 9, 8, 27, 28] is one of the most important components that fuse two equivariant features into one. It provides a

powerful and expressive way [29] to build equivariant networks, while the computational cost of TP is usually considerable. Therefore, there are several directions to accelerate the equivariant networks. First direction is to accelerate TP. eSCN [14, 15, 30] proposes to reduce the SO(3) convolution into SO(2) for TP when one of the inputs of TP is spherical harmonics. Gaunt tensor product [16] makes use of fast spherical Fourier transformation to perform the TP. The other direction is to apply frame averaging (FA) [10, 11, 12], which uses group elements from an equivariant set-valued function called frame to transform the input data and subsequently the model's output, enabling any models to obtain the desired symmetries. Although it is flexible and has no requirement for model architectures, it faces an unsolvable discontinuity problem [11].

## 3 Tensor Decomposition Networks

This section presents the techniques employed in the proposed Tensor Decomposition Network (TDN). In Section 3.1, we introduce a CP-decomposition-based approximation for the tensor product, and in Section 3.2, we detail a path-wise weight-sharing scheme. These strategies effectively reduce both computational cost and parameter count. In Section 3.3, we analyze the computational complexity of the approximate tensor product, and in Section 3.4, we discuss its error bound and universality.

### 3.1 Tensor Product and Its Approximation

The tensor product is a fundamental operation in equivariant neural networks, enabling the coupling of features across multiple vector spaces. However, direct implementation of the tensor product incurs significant computational cost. To mitigate this, we introduce a low-rank approximation using the CANDECOMP/PARAFAC (CP) decomposition to reduce the time complexity.

**Tensor Product.** Before developing our tensor-product approximation we recall the canonical definition of the tensor product in the simplest non-trivial two-order case. The multi-order counterpart and its CP decomposition are introduced in Appendix A. In practice, higher-rank tensors are stored as flattened vectors via a fixed index ordering; this reshaping preserves the vector-space operations. Without loss of generality, we present the following tensor product definition. Let $V_1 = \mathbb{R}^{d_1}$, $V_2 = \mathbb{R}^{d_2}$, and $V_3 = \mathbb{R}^{d_3}$ be finite-dimensional real vector spaces with ordered bases $\{e_i\}_{i=1}^{d_1}$, $\{f_j\}_{j=1}^{d_2}$, $\{g_k\}_{k=1}^{d_3}$. The tensor product $V_1 \otimes V_2$ is the space that corepresents bilinear maps: for every bilinear $m\colon V_1 \times V_2 \to V_3$ there exists a unique linear map $\widetilde{m}\colon V_1 \otimes V_2 \longrightarrow V_3$ such that $m(\boldsymbol{x}, \boldsymbol{y}) = \widetilde{m}(\boldsymbol{x} \otimes \boldsymbol{y})$. With respect to the chosen bases, $\widetilde{m}$ is encoded by a three–way tensor

$$\boldsymbol{M} = \big(\boldsymbol{M}_{kij}\big) \in V_3 \otimes V_1^* \otimes V_2^* \cong \mathbb{R}^{d_3 \times d_1 \times d_2},$$

and $m(\boldsymbol{e}_i, \boldsymbol{f}_j) = \sum_{k=1}^{d_3} \boldsymbol{M}_{kij}\, \boldsymbol{g}_k$. For arbitrary $\boldsymbol{x} = \sum_{i=1}^{d_1} x_i \boldsymbol{e}_i$ and $\boldsymbol{y} = \sum_{j=1}^{d_2} y_j \boldsymbol{f}_j$ we have

$$m(\boldsymbol{x}, \boldsymbol{y}) = \sum_{k=1}^{d_3} \sum_{i=1}^{d_1} \sum_{j=1}^{d_2} \boldsymbol{M}_{kij}\, x_i\, y_j\, \boldsymbol{g}_k = \boldsymbol{M}\big(\boldsymbol{x} \otimes \boldsymbol{y}\big). \tag{2}$$

**CP Decomposition for Tensor Product Approximation.** To reduce time complexity and storage cost of tensor product, we approximate the tensor $\boldsymbol{M}$ via CP decomposition [17] of rank $R$. The CP decomposition writes the tensor product as sum of $R$ rank-1 tensors by decomposing the three-way tensor $\boldsymbol{M}$ such that

$$\boldsymbol{M}_{kij} \approx \sum_{r=1}^{R} \boldsymbol{A}_{kr} \boldsymbol{B}_{ir} \boldsymbol{C}_{jr}, \tag{3}$$

where $\boldsymbol{A} \in \mathbb{R}^{d_3 \times R}$, $\boldsymbol{B} \in \mathbb{R}^{d_1 \times R}$, and $\boldsymbol{C} \in \mathbb{R}^{d_2 \times R}$ are factor matrices capturing the modes of the tensor. Substituting Eq. (3) into Eq. (2) gives

$$m(\boldsymbol{x}, \boldsymbol{y}) \approx \sum_{k=1}^{d_3} \sum_{i=1}^{d_1} \sum_{j=1}^{d_2} \left( \sum_{r=1}^{R} \boldsymbol{A}_{kr} \boldsymbol{B}_{ir} \boldsymbol{C}_{jr} \right) x_i y_j \boldsymbol{g}_k.$$

Then we rearrange the summation to obtain

$$\sum_{r=1}^{R} \left( \sum_{i=1}^{d_1} \boldsymbol{B}_{ir} x_i \right) \left( \sum_{j=1}^{d_2} \boldsymbol{C}_{jr} y_j \right) \left( \sum_{k=1}^{d_3} \boldsymbol{A}_{kr} \boldsymbol{g}_k \right).$$

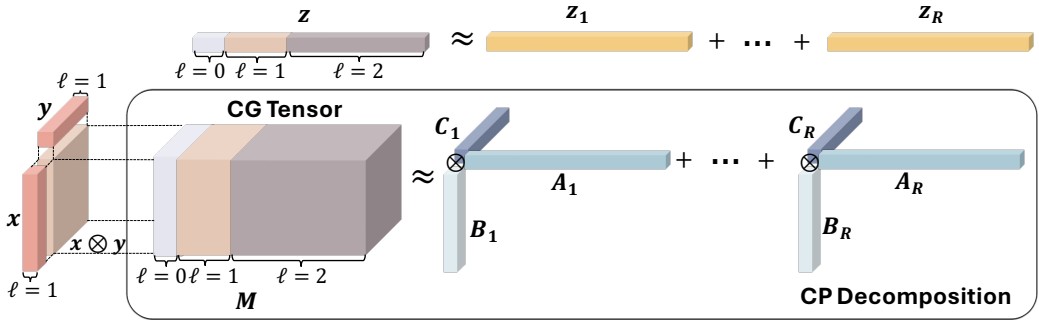

Figure 1: Illustration of approximating the CG tensor product using CP decomposition. The input features $x$ and $y$ both consist of irreps with $\ell = 1$, and the CG tensor product produces output feature $z$ containing irreps with $\ell = 0, 1, 2$.

For clarity, the above approximation for the order-two tensor product $m(x, y)$ can be expressed in matrix form as

$$m(x, y) \approx A\big(B^\top x \odot C^\top y\big), \qquad (4)$$

where $\odot$ denotes the Hadamard product. By the universal property that every bilinear map $m: V_1 \times V_2 \to V_3$ factors uniquely as a linear map $\widetilde{m}: V_1 \otimes V_2 \to V_3$, the tensor product covers important instances used in equivariant learning, notably the CG tensor product [6] and the Gaunt tensor product [16]. Therefore, the tensor product approximation can be employed for those specific cases. In this paper, we primarily discuss CG tensor product approximation.

**CP Decomposition for CG Tensor Product.** Next, we introduce the idea to make use of CP decomposition to accelerate the CG tensor product calculations. Specifically, the CG coefficient tensor is a three-way tensor concatenating CG coefficients $C^{\ell_3}_{\ell_1, \ell_2}$ of all admissible path $(\ell_1, \ell_2, \ell_3)$ such that

$$M = \bigoplus_{\ell_1=0}^{L} \bigoplus_{\ell_2=0}^{L} \bigoplus_{\ell_3=|\ell_1-\ell_2|}^{\min(\ell_1+\ell_2, L)} C^{\ell_3}_{\ell_1, \ell_2}.$$

The key idea of CP decomposition is to break down the CG coefficient tensor $M$ into low-rank matrices. We demonstrate a simple example of CP decomposition for TP with $\ell_1, \ell_2 = 1$ and $\ell_3 \in \{0, 1, 2\}$, as shown in Fig. 1. Eq. (4) allows for the simple batch-wise use

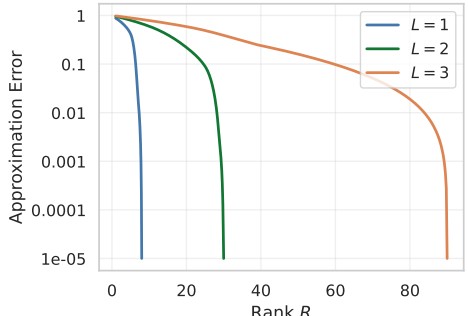

Figure 2: Approximation error for different rank values $R$ across various maximum angular degrees $L$.

of the Hadamard product with a global hyperparameter rank $R$ instead of computing tensor product per path. The higher $R$, the more accurate the tensor product result is. For acceleration, we do not need full rank $R$ and we can use a small $R$ while keeping a low approximation error. We show the rank-error curve in Fig. 2.

## 3.2 Path-Weight Sharing Tensor Product

In equivariant neural networks, the CG tensor product couples features that transform under irreducible representations (irreps) of $SO(3)$. For a maximal degree $L$, a fully connected CG tensor product introduces $\mathcal{O}(L^3)$ paths, each associated with a distinct weight matrix. This leads to a substantial parameter count, which can hinder model efficiency. To mitigate this, we propose a path-weight sharing mechanism to reduce the parameter count while retaining equivariance.

**Concatenating Irreps into a Single Channel.** We first unify the multiplicities across all degrees $\ell$ by setting a common multiplicity $k$. Letting $x^{(\ell)} \in \mathbb{R}^{k \times (2\ell+1)}$ denote the multiplicity-$k$ irrep of degree $\ell$, we concatenate the irrep axes into a single channel:

$$\tilde{x} = \text{concat}_{\ell=0}^{L} x^{(\ell)} \in \mathbb{R}^{k \times (L+1)^2}.$$

All irreps now share a common multiplicity index within, so the linear projection and CG contraction over irreps features are executed as batched matrix operations whose operands reside contiguously in memory; this yields coalesced memory access on GPUs and improved cache locality on CPUs.

**Path-Weight Sharing Tensor Product.** We further reduce the parameter count by applying a path-weight sharing scheme. Let $\boldsymbol{W}_{\ell_1,\ell_2}^{\ell_3}$ denote the multiplicity-space weight matrix associated with the path $(\ell_1, \ell_2, \ell_3)$, where $0 \leq \ell_1, \ell_2, \ell_3 \leq L$ and $c$ is the parameter count of weight of each path. Instead of assigning a unique matrix for each path, we set all such matrices equal to a single learnable parameter $\boldsymbol{W}$, i.e. $\boldsymbol{W}_{\ell_1,\ell_2}^{\ell_3} \equiv \boldsymbol{W}$ for every admissible path $(\ell_1, \ell_2, \ell_3)$. This collapses the $\mathcal{O}(L^3)$ distinct weight tensors of the naïve implementation into one, reducing the parameter count from $\mathcal{O}(cL^3)$ to $\mathcal{O}(c)$. Because the sharing operates exclusively on multiplicity indices, the irrep content of each block and hence full SO(3) equivariance is retained. The resulting layer therefore retains the classic CG tensor product structure while offering an order-of-magnitude reduction in parameters. We also extend this scheme to equivariant linear layers and use it in our main experiments.

### 3.3 Complexity Analysis of Approximate Tensor Product

In this section, we analyze the computational complexity of the proposed approximate tensor product using CP decomposition. We first investigate the rank of the tensor product. Let $\mathrm{rank}_{\mathrm{CP}}(\boldsymbol{M})$ denote the *CP rank* of the three-way tensor $\boldsymbol{M} \in \mathbb{R}^{d_3 \times d_1 \times d_2}$, representing the minimal rank $R$ such that there is an equality for Eq. (3), i.e.

$$\mathrm{rank}_{\mathrm{CP}}(\boldsymbol{M}) = \min\Big\{ R \in \mathbb{N}^+ \ \Big| \ \boldsymbol{M}_{kij} = \sum_{r=1}^{R} \boldsymbol{A}_{kr}\boldsymbol{B}_{ir}\boldsymbol{C}_{jr} \Big\}.$$

Determining $\mathrm{rank}_{\mathrm{CP}}(\boldsymbol{M})$ exactly is NP-hard [17]. In practice one specifies a rank $R$ as a hyper-parameter and optimizes the factor matrices $\boldsymbol{A} \in \mathbb{R}^{d_3 \times R}$, $\boldsymbol{B} \in \mathbb{R}^{d_1 \times R}$, $\boldsymbol{C} \in \mathbb{R}^{d_2 \times R}$ to minimize a prescribed loss. A generic upper bound $\mathrm{rank}_{\mathrm{CP}}(\boldsymbol{M}) \leq \min\{d_1 d_2,\ d_1 d_3,\ d_2 d_3\}$ limits the choices of $R$. Consequently, algorithmic pipelines treat $R$ as an external choice, balancing approximation accuracy against computational cost.

By using CP decomposition, the computational cost for evaluating the approximate tensor product in Eq. (4) reduces to $\mathcal{O}\big(R(d_1 + d_2 + d_3)\big)$, a significant reduction compared to the full tensor contraction cost of $\mathcal{O}(d_1 d_2 d_3)$ using Eq. (2). Similarly, the space complexity decreases from $\mathcal{O}(d_1 d_2 d_3)$ to $\mathcal{O}(R(d_1 + d_2 + d_3))$, which provides substantial savings when $R$ is small.

For the approximation of the CG tensor product, where $d_1, d_2, d_3 \propto \mathcal{O}(L^2)$, the computational complexity further reduces to $\mathcal{O}(RL^2)$. In our experiments, we select $R = 7L^2$. The error curve for the CP decomposition with varying $R$ is discussed in Section 4.1.

### 3.4 Error Bound and Universality Analysis

The error bound and universality analysis provide theoretical guarantees for the CP decomposition of the tensor product in the proposed Tensor Decomposition Network (TDN). This section establishes the error bound for both the approximation and equivariance, demonstrating how the approximation error depends on the spectral tail of the tensor's singular values. Additionally, we formalize the universality property of the CP decomposition, showing that as the rank $R$ increases, the CP-decomposition-based tensor product can accurately approximate any SO(3)-equivariant bilinear map, thereby preserving the expressive power of the tensor product while reducing computational complexity.

**Error Bound of Approximate Tensor Product.** Given a rank $R \leq \mathrm{rank}_{\mathrm{CP}}(\boldsymbol{M})$, one seeks CP-decomposition-based approximation $\widehat{\boldsymbol{M}}$ by minimizing $\|\boldsymbol{M} - \widehat{\boldsymbol{M}}\|$ in a chosen norm, typically the Frobenius norm. Although the non-convex optimization may admit spurious local minima, the optimization of CP decomposition possesses an essentially unique rank-$R$ approximation under Kruskal's condition [17], and modern alternating least-squares or gradient methods converge to it under mild coherence assumptions [31]. For error estimation in our setting, we specialize to the CG tensor product, where we let all irreps have the same maximum degree and all dimensions equal such that $d = d_1 = d_2 = d_3$. Given the singular values $\sigma_k^{(n)}$ of the mode-$n$ matricization $\boldsymbol{M}_{(n)}$ of $\boldsymbol{M}$ and

truncating each $M_{(n)}$ to rank $R_T = \lfloor R^{1/3} \rfloor$, a priori approximation error bound [32] gives

$$\left\| M - M_{\text{truncated}} \right\|_F \leq \Big( \sum_{n=1}^{3} \sum_{k>R_T} \sigma_k^{(n)2} \Big)^{1/2}.$$

To quantify the loss of SO(3)-equivariance incurred by CP decomposition, let $R \in \text{SO}(3)$ with representation the Wigner D-matrix $D(R)$, and the SO(3)-equivariance error is estimated by

$$\varepsilon(R, x, y) = \| \widehat{M}(D(R)x \otimes D(R)y) - D(R)\widehat{M}(x \otimes y) \|.$$

The following result bounds this error uniformly over SO(3) with the proof in Appendix B.1.

**Theorem 3.1 (Equivariance Error Bound of CP Decomposition):** *Let CG tensor $M \in \mathbb{R}^{d \times d \times d}$ and $\widehat{M}$ be the rank-$R$ CP-decomposition-based approximation obtained by Frobenius minimization. For any rotation $R \in \text{SO}(3)$ and any bounded representations $x, y \in \mathbb{R}^d$, $\|x\|, \|y\| \leq C$, we have*

$$\varepsilon(R, x, y) \leq 2C^2 \Big( \sum_{n=1}^{3} \sum_{k>R_T} \sigma_k^{(n)2} \Big)^{1/2},$$

*where $R_T = \lfloor R^{1/3} \rfloor$ and $\sigma_k^{(n)}$ is the $k$-th singular value of mode-$n$ matricization of $M$.*

Empirical estimates of both the approximation and equivariance errors are provided in Section 4.1.

**Universality of Approximate Tensor Product.** Because the tensor product is universal for bilinear maps, any SO(3)-equivariant bilinear operator can be expressed as a composition of a tensor product followed by a suitable $M$ onto an equivariant subspace. Consequently, the CP-decomposition-based approximation developed above inherits this universality: for every equivariant tensor product there exists a rank-$R$ approximation that converges to it as $R \to \text{rank}_{\text{CP}}(M)$. The following theorem formalizes the expressivity of our approximation scheme with the proof in Appendix B.2.

**Theorem 3.2 (Universality of CP Decomposition):** *Consider* SO(3)-*representations $x \in V_1 \cong \mathbb{R}^{d_1}$, $y \in V_2 \cong \mathbb{R}^{d_2}$ and co-domain $V_3 \cong \mathbb{R}^{d_3}$. For any* SO(3)-*equivariant bilinear map $\Phi$, there exist $B \in \mathbb{R}^{d_1 \times R}$, $C \in \mathbb{R}^{d_2 \times R}$, $A \in \mathbb{R}^{d_3 \times R}$ such that $\Phi$ can be written as*

$$\Phi(x, y) = A(B^\top x \odot C^\top y) \in V_3,$$

*with $R \leq d_1 d_2$.*

## 4 Experiments

The effectiveness of our method is assessed on two benchmarks: the PubChemQCR dataset, which contains millions of molecular relaxation snapshots, and the established Open Catalyst Project (OCP) datasets. Section 4.1 describes both benchmarks and model configurations. Section 4.2 compares our model with several baselines on PubChemQCR and its subset PubChemQCR-S, while Section 4.3 and Section 4.4 report results of our model compared to several tensor-product baselines on OC20 and OC22 from OCP, respectively. Section 4.5 then analyses the computational efficiency of our approach as the maximum angular degree in the tensor product is varied. Section 4.6 further reports ablations of the proposed components across our model and a second widely used architecture.

### 4.1 Experimental Setup

**Datasets.** We use a newly curated dataset PubChemQCR [36], comprising high-fidelity molecular trajectories derived from the PubChemQC database [37]. This dataset encompasses a diverse range of molecular systems, capturing potential energy surfaces and force information critical for understanding molecular interactions. The full dataset consists of 3,471,000 trajectories and 105,494,671 DFT-calculated molecular snapshots, with each snapshot containing molecular structure, total energy, and forces. For training efficiency, we use the smaller subset, PubChemQCR-S, comprising 40,979 trajectories and 1,504,431 molecular snapshots for model benchmarking. The

Table 1: Comparison of model performance on energy and force predictions for the PubChemQCR and the PubChemQCR-S dataset. Our model is trained to compare against several baseline methods on the PubChemQCR-S dataset, including SchNet [18], PaiNN [4], MACE [8], PACE [33], FAENet [10], NequIP [9], SevenNet [34], Allegro [35], and Equiformer [7]. On the full PubChemQCR dataset, we compare our model with SchNet [18] and PaiNN [4]. The best results are shown in **bold**.

| | | VALIDATION | | TEST | |
| DATASET | MODEL | ENERGY MAE (meV/atom) ↓ | FORCE RMSE (meV/Å) ↓ | ENERGY MAE (meV/atom) ↓ | FORCE RMSE (meV/Å) ↓ |
|---|---|---|---|---|---|
| SMALL SUBSET | SCHNET | 5.30 | 56.55 | 5.55 | 56.22 |
| | PAINN | 5.13 | 46.34 | 5.33 | 46.92 |
| | NEQUIP | 7.37 | 54.78 | 8.27 | 55.59 |
| | SEVENNET | 8.77 | 47.63 | 10.21 | 48.05 |
| | ALLEGRO | 10.86 | 60.71 | 10.80 | 61.44 |
| | FAENET | 7.28 | 60.24 | 8.70 | 60.51 |
| | MACE | 7.54 | 51.46 | 7.47 | 45.70 |
| | PACE | 6.24 | 50.54 | 6.53 | 51.43 |
| | EQUIFORMER | 4.69 | 34.67 | 5.38 | 35.11 |
| | TDN | **4.46** | **26.94** | **5.01** | **26.43** |
| FULL | SCHNET | 7.14 | 65.22 | 7.71 | 67.38 |
| | PAINN | 3.62 | 38.30 | 3.49 | 39.28 |
| | TDN | **1.65** | **19.46** | **1.50** | **20.44** |

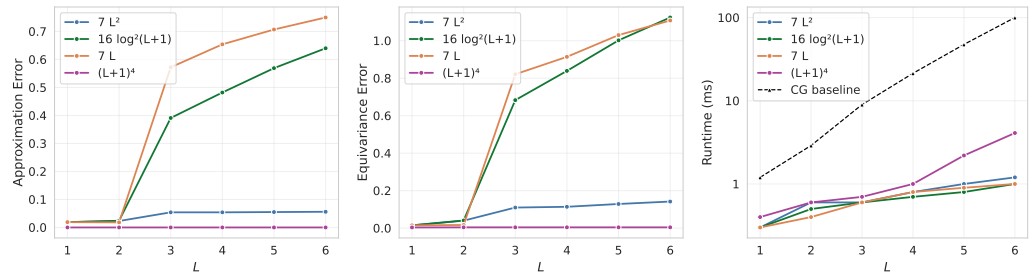

(a) Approximation error vs. $L$    (b) SO(3)-equivariance error vs. $L$    (c) Runtime vs. $L$ (log–scale)

Figure 3: Scaling behaviours of the CP-decomposition-based tensor product under different maximum angular degree schedules: (a) approximation error, (b) SO(3)-equivariance error, and (c) runtime. The error and runtime of the CP decomposition-based tensor product depend on the chosen rank, multiplicities, and the maximum angular degree, and are not tied to a specific dataset.

subset is split into training, validation, and testing sets using a 60%-20%-20% ratio, while the full dataset employs an 80%-10%-10% split to assess generalizability. To prevent data leakage, each trajectory is assigned to only one split in both PubChemQCR and PubChemQCR-S.

The second dataset used in this study is the Open Catalyst Project (OCP) [38], which publishes extensive open datasets of DFT relaxations for adsorbate–catalyst surfaces and hosts public leaderboard challenges. The flagship releases, OC20 [39] and OC22 [40], encompass thousands of chemical compositions, crystal facets, and adsorbates, serving as comprehensive benchmarks for surrogate models aiming to replace computationally intensive calculations. Each release defines multiple tasks, including *Initial-Structure-to-Relaxed-Energy* (IS2RE), which require accurate predictions of total energies, per-atom forces, and relaxed geometries. We conduct experiments on OC20 IS2RE and OC22 IS2RE, using the data splits and configurations specified in the official OCP repository.

**CP Rank Selection.** We approximate the fully connected CG tensor product using the rank-$R$ CP decomposition across various configurations of the maximum angular degree $L$. Four rank schedules are evaluated: $R = (L + 1)^4$, $R = 16 \log^2(L + 1)$, $R = 7L$, and $R = 7L^2$. For each configuration we measure (i) **Approximation Error:** The relative CP tensor product error, calculated as $\|M(x \otimes y) - \widehat{M}(x \otimes y)\|_F / \|M(x \otimes y)\|_F$; (ii) **Equivariance Error:** The expected SO(3)-equivariance error, defined as $\mathbb{E}_{R,x,y}[\varepsilon(R, x, y)]$, where the expectation is averaged over

1000 random rotations $R$ and random vectors $x, y$; and (iii) **Execution Time:** The runtime for the CP-decomposition-based tensor product under each rank schedule. Among the tested schedules, the quadratic schedule $R = 7L^2$ consistently achieves the best accuracy-efficiency trade-off, while also requiring significantly less computation time compared to the CG tensor product baseline. Thus, this schedule provides the best balance between accuracy and computational cost and is adopted for all subsequent experiments. Further rank ablations are described in Appendix D.

**Model Design.** Building on the capabilities of graph transformers, we develop the *Tensor-Decomposition Network* (TDN) by modifying the publicly available Equiformer architecture [7]. Specifically, we replace every channel-wise linear projection, normalization layer, and activation in Equiformer with batched matrix operations that act simultaneously on the multiplicity dimension of each irrep block, eliminating the need for costly slicing and reshaping between tensor and vector representations. Additionally, the depth-wise tensor product mechanism is removed and the core CG tensor products within the self-attention mechanism are substituted with our rank-$R$ CP-decomposition-based tensor product from Section 3.1, integrated with the path-weight-sharing scheme from Section 3.2. This design preserves the expressive power of CG tensor product while significantly improving memory and computational efficiency. Note that for each maximum degree $L$, we precompute a rank-$R$ CP decomposition of the CG coefficient tensor once and cache the factors. These factors are treated as constants during all training runs. We further integrate the equivariant linear layer with the path-weight-sharing scheme. For the OCP dataset, the same model architecture is applied and a subset of experiments additionally incorporate initial node embeddings following [25]. The detailed model configurations for TDN are described in Appendix C.

**Baseline Implementations.** For the PubChemQCR benchmark, we reimplement each baseline model based on its official repository. Hyperparameters are adopted from the best configurations reported in the original papers or, if unspecified, are tuned using the PubChemQCR-S dataset. All model configurations of baseline models are described in Appendix C.

## 4.2 Results on PubChemQCR

We evaluate our method on both PubChemQCR-S and the full PubChemQCR dataset. For the small split we compare against nine state-of-the-art models: SchNet [18], PaiNN [4], MACE [8], PACE [33], FAENet [10], NequIP [9], SevenNet [34], Allegro [35], and Equiformer [7]. On the full PubChemQCR dataset, we benchmark against SchNet and PaiNN, the only baselines that scale to its size within our hardware budget. Performance is reported as mean absolute error (MAE) for energies and root-mean-square error (RMSE) for forces over the validation and testing splits. Note that all results are selected based on the lowest validation energy error. The results are shown in Table 1, the proposed TDN model achieves the lowest energy and force prediction errors across PubChemQCR-S and the full PubChemQCR dataset, outperforming all baseline methods. In addition, the performance of TDN improves further as the size of the dataset increases.

**Training Setup.** Across both the PubChemQCR and PubChemQCR-S benchmarks, we adopt a uniform training protocol: a cutoff radius of 4.5 Å; the Adam optimizer with an initial learning rate of $5 \times 10^{-4}$; and a REDUCELRONPLATEAU scheduler with a patience of 2 epochs. All models are trained for up to 100 epochs on PubChemQCR-S and up to 15 epochs on the full PubChemQCR dataset. Unless otherwise noted, experiments are executed on NVIDIA A100-80GB GPUs.

## 4.3 Results on OC20

Table 2 summarizes our performance on the principal OC20 task IS2RE-DIRECT, which predicts the relaxed adsorption energy directly from the initial geometry (no noisy-node auxiliary loss). We benchmark against the widely-used baselines reported to date, including SchNet [18], DimeNet++ [41], GemNet-dT [19], SphereNet [20], Equiformer [7] and EquiformerV2 [15]. Metrics follow the official OC20 protocol: energy mean absolute error (MAE, eV) and energy within threshold (EwT, %) in IS2RE-DIRECT for four validation sub-splits: distribution adsorbates and catalysts (ID), out-of-distribution adsorbates (OOD-Ads), out-of-distribution catalysts (OOD-Cat), and out-of-distribution adsorbates and catalysts (OOD-Both). The IS2RE-DIRECT results are presented in Table 2, demonstrating that our model achieves performance comparable to Equiformer while being more efficient,

Table 2: Comparison of model performance on energy predictions for OC20 IS2RE-DIRECT validation set without noisy-node auxiliary loss. Our model is trained to compare against several baseline methods, including SchNet [18], DimeNet++ [41], GemNet-dT [19], SphereNet [20], Equiformer [7] and EquiformerV2 [15]. The best results are shown in **bold** and the second best results are shown with underlines.

| MODEL | ENERGY MAE (eV) ↓ | | | | | EwT (%) ↑ | | | | |
| | ID | OOD ADS | OOD CAT | OOD BOTH | AVERAGE | ID | OOD ADS | OOD CAT | OOD BOTH | AVERAGE |
| --- | --- | --- | --- | --- | --- | --- | --- | --- | --- | --- |
| SCHNET | 0.6465 | 0.7074 | 0.6475 | 0.6626 | 0.6660 | 2.96 | 2.22 | 3.03 | 2.38 | 2.65 |
| DIMENET++ | 0.5636 | 0.7127 | 0.5612 | 0.6492 | 0.6217 | 4.25 | 2.48 | 4.40 | 2.56 | 3.42 |
| GEMNET-dT | 0.5561 | 0.7342 | 0.5659 | 0.6964 | 0.6382 | 4.51 | 2.24 | 4.37 | 2.38 | 3.38 |
| SPHERENET | 0.5632 | 0.6682 | 0.5590 | 0.6190 | 0.6024 | 4.56 | 2.70 | 4.59 | 2.70 | 3.64 |
| EQUIFORMER | 0.5088 | **0.6271** | **0.5051** | **0.5545** | **0.5489** | 4.88 | **2.93** | 4.92 | **2.98** | 3.93 |
| EQUIFORMERV2 | 0.5161 | 0.7041 | 0.5245 | 0.6365 | 0.5953 | - | - | - | - | - |
| TDN | **0.5085** | 0.6668 | 0.5104 | 0.5875 | 0.5683 | **5.21** | 2.54 | **5.04** | **2.98** | **3.94** |

as detailed in Section 4.5. This highlights the effectiveness of our architecture in maintaining accuracy while significantly reducing computational costs.

**Training Setup.** For the IS2RE-DIRECT task, we follow Equiformer's optimization setup [7] by a maximum angular degree $L = 1$, an AdamW optimizer with a learning rate of $2 \times 10^{-4}$ and a weight decay of $10^{-3}$, a batch size of 32, and a cosine-decay learning-rate schedule. A warm-up is employed for 2 epochs on IS2RE-DIRECT, with a warm-up factor of 0.2; the cosine decay then runs over 30 training epochs. IS2RE-DIRECT experiments are run on a single NVIDIA RTX A6000-48GB GPU.

## 4.4 Results on OC22

Following the OC20 evaluation protocol, we evaluate on the OC22 IS2RE-DIRECT test set, which predicts relaxed energy directly from the initial structure and omits the noisy-node auxiliary loss. We benchmark against strong baselines up to date as listed in Table 3 and report mean absolute error (MAE, eV) for the in-distribution (ID) and out-of-distribution (OOD) splits by averaging across the four standard OC22 sub-splits using the same split scheme as OC20 IS2RE-DIRECT task. As summarized in Table 3, TDN achieves the lowest ID MAE and the best overall average MAE, and it attains the second-best OOD MAE, validating the high effectiveness of TDN.

Table 3: Comparison of model performance on energy predictions for OC22 IS2RE-DIRECT testing set. We compare with several baseline methods, including SchNet [18], DimeNet++ [41], PaiNN [4], GemNet-dT [19], and coGN [42]. The best results are shown in **bold** and the second best results are shown with underlines.

| MODEL | MAE (ID) (eV) | MAE (OOD) (eV) | AVERAGE (eV) |
| --- | --- | --- | --- |
| SCHNET | 2.00 | 4.85 | 3.42 |
| DIMENET++ | 1.96 | 3.52 | 2.74 |
| PAINN | 1.72 | 3.68 | 2.70 |
| GEMNET-dT | 1.68 | 3.08 | 2.38 |
| COGN | 1.62 | **2.81** | 2.21 |
| TDN | **1.49** | 2.92 | **2.20** |

**Training Setup.** On OC22 IS2RE, we use the same optimization setup as OC20 by a maximum angular degree $L = 1$, an AdamW optimizer with a learning rate of $2 \times 10^{-4}$, a weight decay of $10^{-3}$, a batch size of 32, and a cosine learning-rate decay with a 2-epoch warm-up for 1000 epochs on a single NVIDIA A100-80GB GPU.

## 4.5 Efficiency of CP-Decomposition-Based Tensor Product

To evaluate the speed-up achieved by our CP-decomposition-based tensor product in Section 3.1, we benchmark its inference runtime against the fully connected CG tensor product implementation in `e3nn`. As shown in Fig. 3c, the proposed approximation accelerates by factors of $4.0\times$, $4.8\times$, $15.0\times$, $26.7\times$, $47.6\times$, and $83.6\times$ for maximum degrees $L = 1, 2, 3, 4, 5$, and $6$, respectively.

Table 4: Throughput and parameter count comparison between TDN and Equiformer across different values of maximum degree $L$.

| MODEL | THROUGHPUT (samples/sec) | PARAMETERS |
| --- | --- | --- |
| EQUIFORMER (L=1) | 311.7 | 12.1M |
| TDN (L=1) | 770.8 (× **2.47**) | 4.5M (× **0.37**) |
| EQUIFORMER (L=2) | 71.9 | 27.9M |
| TDN (L=2) | 312.4 (× **4.34**) | 4.5M (× **0.16**) |
| EQUIFORMER (L=3) | 26.1 | 54.7M |
| TDN (L=3) | 220.4 (× **8.44**) | 4.5M (× **0.08**) |

Since TDN is derived from Equiformer by replacing each CG block with CP decomposition and incorporating path-weight sharing, and removing depth-wise tensor product mechanism, we further benchmark end-to-end throughput and parameter count for both networks on a single NVIDIA A100-80GB GPU and Xeon Gold 6258R processor with a batch size of 128; detailed model configurations are provided in Appendix D. The results in Table 4 show that TDN processes $2.47\times$ to $8.44\times$ more structures per second and uses $63\%$–$92\%$ fewer parameters than Equiformer. As the maximum degree $L$ increases, TDN achieves higher throughput and requires fewer parameters. A comprehensive time ablation of CP decomposition and path-weight sharing mechanism is provided in Table 7.

### 4.6 Efficiency Ablation of CP-Decomposition-Based Tensor Product

To further evaluate the efficiency contributions of our proposed components, we perform ablations on the PubChemQCR-S dataset by removing the path-weight sharing tensor product and the CP decomposition of TDN while holding all other components fixed. As shown in Section 4.6, starting from TDN, enabling both mechanisms reduces training time by roughly 78%. TDN is trained with a maximum degree $L = 2$, four graph transformer layers, a hidden dimension of 64. We also evaluate NequIP [9]

Table 5: Efficiency ablation of the path-weight sharing tensor product (PS) and CP-decomposition-based tensor product (CP). Results are shown for TDN and NequIP [9] on PubChemQCR-S dataset.

| MODEL | TRAINING TIME (min/epoch) |
|---|---|
| TDN *w/o CP + PS* | 19.0 |
| TDN | **4.2** ($\times 0.22$) |
| NEQUIP | 7.5 |
| NEQUIP + *CP +PS* | **2.0** ($\times 0.26$) |

by augmenting the base model with path-weight sharing tensor product and CP decomposition. This cuts computational cost nearly 74%. NequIP is trained with a maximum degree $L = 2$, four interaction blocks, and multiplicity 64. All experiments are conducted on a single NVIDIA RTX A6000-48GB GPU. A full table of ablation studies including accuracy results is presented in Table 9.

## 5 Limitations

Our approach accelerates $SO(3)$-equivariant tensor products by replacing the full CG tensor product with a low-rank CP decomposition. We substantiate both its theoretical speed-up and its empirical efficacy across multiple benchmarks. One limitation of our approach is rank selection: the minimal CP rank needed to attain a given approximation error is unknown in general, and computing it exactly is NP-hard [17]. We therefore employ an empirical rank scheduler; however, this scheduler is tailored to the CG tensor and must be re-derived when extending to other equivariant tensor products. A second limitation is that path-weight sharing mechanism, while reducing parameters and memory, can introduce a small accuracy drop. In future work, we will (i) study broader families of group-equivariant tensor products, e.g., $SO(2)$ convolution [14], Gaunt tensor products [16], or more complex tensor products [43], to characterize their optimal rank profiles and develop general rank-selection criteria and adaptive rank selector, and (ii) design adaptive path-selection and grouping strategies that preserve the efficiency benefits of weight sharing while recovering performance.

## 6 Summary

In this work, we present Tensor Decomposition Networks (TDNs), a novel framework designed to accelerate the computationally intensive Clebsch-Gordan (CG) tensor product in $SO(3)$-equivariant networks through low-rank tensor decomposition. By leveraging CANDECOMP/PARAFAC (CP) decomposition and implementing path-weight sharing mechanism, TDNs effectively reduce both parameter count and computational complexity while preserving the expressive power of conventional CG tensor products. We also analyze time complexity, derive approximation error bounds, and establish universality of our approach, providing theoretical guarantees. Extensive evaluations on a newly curated PubChemQCR dataset and commonly used OC20 and OC22 benchmarks demonstrate that TDNs achieve comparable predictive accuracy to state-of-the-art models while significantly reducing runtime. The proposed framework provides a plug-and-play alternative to conventional CG tensor products, making it a promising approach for large-scale molecular simulations.

## Acknowledgments

SJ acknowledges support from ARPA-H under grant 1AY1AX000053, National Institutes of Health under grant U01AG070112, and National Science Foundation under grant IIS-2243850. XQ acknowledges support from the Air Force Office of Scientific Research (AFOSR) under grant FA9550-24-1-0207. We acknowledge Lambda, Inc. and NVIDIA for providing the computational resources for this project.

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

## A  Multilinear Maps and Tensor Decomposition in Arbitrary Order

**Tensor product in arbitrary order.**  Let $N \geq 2$ and let $V_i = \mathbb{R}^{d_i}$ for $i = 1, \ldots, N+1$ be finite-dimensional real vector spaces equipped with fixed ordered bases $\{e_k^{(i)}\}_{k=1}^{d_i}$. A multilinear map

$$m \colon V_1 \times \cdots \times V_N \longrightarrow V_{N+1}$$

is uniquely represented by a linear map

$$\widetilde{m} \colon V_1 \otimes \cdots \otimes V_N \longrightarrow V_{N+1}, \qquad m(\boldsymbol{x}^{(1)}, \ldots, \boldsymbol{x}^{(N)}) = \widetilde{m}(\boldsymbol{x}^{(1)} \otimes \cdots \otimes \boldsymbol{x}^{(N)}).$$

With respect to the chosen bases, $\widetilde{m}$ is encoded by an $(N+1)$-way tensor

$$\boldsymbol{M} \in V_{N+1} \otimes V_1^* \otimes \cdots \otimes V_N^* \cong \mathbb{R}^{d_{N+1} \times d_1 \times \cdots \times d_N},$$

defined through

$$m(\boldsymbol{x}^{(1)}, \ldots, \boldsymbol{x}^{(N)}) \;=\; \sum_{i_{N+1}=1}^{d_{N+1}} \sum_{i_1=1}^{d_1} \cdots \sum_{i_N=1}^{d_N} \boldsymbol{M}_{i_{N+1} i_1 \cdots i_N} x_{i_1}^{(1)} \cdots x_{i_N}^{(N)} \boldsymbol{e}_{i_{N+1}}^{(N+1)}. \qquad (5)$$

**CP decomposition in arbitrary order.**  To lower computational cost, we can approximate $\boldsymbol{M}$ by a rank-$R$ CP decomposition

$$\boldsymbol{M}_{i_{N+1} i_1 \cdots i_N} \;\approx\; \sum_{r=1}^{R} \boldsymbol{A}_{i_{N+1} r} \, \boldsymbol{B}_{i_1 r}^{(1)} \cdots \boldsymbol{B}_{i_N r}^{(N)},$$

where $\boldsymbol{A} \in \mathbb{R}^{d_{N+1} \times R}$ and $\boldsymbol{B}^{(i)} \in \mathbb{R}^{d_i \times R}$ $(i = 1, \ldots, N)$. Substituting into Eq. (5) and rearranging the summation the we obtain

$$m(\boldsymbol{x}^{(1)}, \ldots, \boldsymbol{x}^{(N)}) \;\approx\; \sum_{r=1}^{R} \left( \sum_{i_1=1}^{d_1} \boldsymbol{B}_{i_1 r}^{(1)} x_{i_1}^{(1)} \right) \cdots \left( \sum_{i_N=1}^{d_N} \boldsymbol{B}_{i_N r}^{(N)} x_{i_N}^{(N)} \right) \left( \sum_{i_{N+1}=1}^{d_{N+1}} \boldsymbol{A}_{i_{N+1} r} \boldsymbol{e}_{i_{N+1}}^{(N+1)} \right),$$

which can be expressed in matrix form

$$m(\boldsymbol{x}^{(1)}, \ldots, \boldsymbol{x}^{(N)}) \;\approx\; \boldsymbol{A} \left( \boldsymbol{B}^{(1)\top} \boldsymbol{x}^{(1)} \odot \cdots \odot \boldsymbol{B}^{(N)\top} \boldsymbol{x}^{(N)} \right)$$

where $\odot$ denotes the Hadamard product.

## B  CP-Based Tensor Product

### B.1  Error Bound of CP-Based Tensor Product

**Theorem B.1 (Equivariance Error Bound of CP Decomposition):** *Let CG tensor $\boldsymbol{M} \in \mathbb{R}^{d \times d \times d}$ and $\widehat{\boldsymbol{M}}$ be the rank-$R$ CP-decomposition-based approximation obtained by Frobenius minimization. For any rotation $\boldsymbol{R} \in SO(3)$ and any bounded representations $\boldsymbol{x}, \boldsymbol{y} \in \mathbb{R}^d, \|\boldsymbol{x}\|, \|\boldsymbol{y}\| \leq C$, we have*

$$\varepsilon(\boldsymbol{R}, \boldsymbol{x}, \boldsymbol{y}) \;\leq\; 2C^2 \Big( \sum_{n=1}^{3} \sum_{k > R_T} \sigma_k^{(n)2} \Big)^{1/2},$$

*where $R_T = \lfloor R^{1/3} \rfloor$ and $\sigma_k^{(n)}$ is the $k$-th singular value of mode-$n$ matricization of $\boldsymbol{M}$.*

*Proof.*  Given any rotation $\boldsymbol{R}$ and $SO(3)$-representations $\boldsymbol{x}, \boldsymbol{y}$,

$$\|D(\boldsymbol{R})\boldsymbol{x}\| = \sqrt{\boldsymbol{x}^\top D(\boldsymbol{R})^* D(\boldsymbol{R}) \boldsymbol{x}} = \sqrt{\boldsymbol{x}^\top \boldsymbol{I} \boldsymbol{x}} = \sqrt{\boldsymbol{x}^\top \boldsymbol{x}} = \|\boldsymbol{x}\|,$$

and

$$\|(D(\boldsymbol{R}) \otimes D(\boldsymbol{R}))(\boldsymbol{x} \otimes \boldsymbol{y})\|_F = \sqrt{\mathrm{Tr}((\boldsymbol{x} \otimes \boldsymbol{y})^\top (D(\boldsymbol{R}) \otimes D(\boldsymbol{R}))^* (D(\boldsymbol{R}) \otimes D(\boldsymbol{R}))(\boldsymbol{x} \otimes \boldsymbol{y}))}$$
$$= \sqrt{\mathrm{Tr}((\boldsymbol{x} \otimes \boldsymbol{y})^\top (D(\boldsymbol{R})^* D(\boldsymbol{R}) \otimes D(\boldsymbol{R})^* D(\boldsymbol{R}))(\boldsymbol{x} \otimes \boldsymbol{y}))}$$
$$= \sqrt{\mathrm{Tr}((\boldsymbol{x} \otimes \boldsymbol{y})^\top (\boldsymbol{I} \otimes \boldsymbol{I})(\boldsymbol{x} \otimes \boldsymbol{y}))}$$
$$= \sqrt{\mathrm{Tr}((\boldsymbol{x} \otimes \boldsymbol{y})^\top (\boldsymbol{x} \otimes \boldsymbol{y}))}$$
$$= \|\boldsymbol{x} \otimes \boldsymbol{y}\|_F \tag{6}$$

Since the tensor product Frobenius norm is equal to multiplication of Frobenius norms, we have

$$\varepsilon(\boldsymbol{R}, \boldsymbol{x}, \boldsymbol{y}) = \|\widehat{\boldsymbol{M}}(D(\boldsymbol{R})\boldsymbol{x} \otimes D(\boldsymbol{R})\boldsymbol{y}) - D(\boldsymbol{R})\widehat{\boldsymbol{M}}(\boldsymbol{x} \otimes \boldsymbol{y})\|$$
$$= \|\widehat{\boldsymbol{M}}(D(\boldsymbol{R})\boldsymbol{x} \otimes D(\boldsymbol{R})\boldsymbol{y}) - \boldsymbol{M}(D(\boldsymbol{R})\boldsymbol{x} \otimes D(\boldsymbol{R})\boldsymbol{y})$$
$$+ D(\boldsymbol{R})\boldsymbol{M}(\boldsymbol{x} \otimes \boldsymbol{y}) - D(\boldsymbol{R})\widehat{\boldsymbol{M}}(\boldsymbol{x} \otimes \boldsymbol{y})\|$$
$$\leq \|\widehat{\boldsymbol{M}}(D(\boldsymbol{R})\boldsymbol{x} \otimes D(\boldsymbol{R})\boldsymbol{y}) - \boldsymbol{M}(D(\boldsymbol{R})\boldsymbol{x} \otimes D(\boldsymbol{R})\boldsymbol{y})\|$$
$$+ \|D(\boldsymbol{R})\boldsymbol{M}(\boldsymbol{x} \otimes \boldsymbol{y}) - D(\boldsymbol{R})\widehat{\boldsymbol{M}}(\boldsymbol{x} \otimes \boldsymbol{y})\|$$
$$= \|(\widehat{\boldsymbol{M}} - \boldsymbol{M})(D(\boldsymbol{R})\boldsymbol{x} \otimes D(\boldsymbol{R})\boldsymbol{y})\|$$
$$+ \|D(\boldsymbol{R})(\boldsymbol{M} - \widehat{\boldsymbol{M}})(\boldsymbol{x} \otimes \boldsymbol{y})\| \tag{7}$$
$$= \|(\widehat{\boldsymbol{M}} - \boldsymbol{M})(D(\boldsymbol{R}) \otimes D(\boldsymbol{R}))(\boldsymbol{x} \otimes \boldsymbol{y})\|$$
$$+ \|D(\boldsymbol{R})(\boldsymbol{M} - \widehat{\boldsymbol{M}})(\boldsymbol{x} \otimes \boldsymbol{y})\|$$
$$\leq \|\widehat{\boldsymbol{M}} - \boldsymbol{M}\|_F \|(D(\boldsymbol{R}) \otimes D(\boldsymbol{R}))(\boldsymbol{x} \otimes \boldsymbol{y})\|_F$$
$$+ \|\boldsymbol{M} - \widehat{\boldsymbol{M}}\|_F \|\boldsymbol{x} \otimes \boldsymbol{y}\|_F$$
$$= 2\|\widehat{\boldsymbol{M}} - \boldsymbol{M}\|_F \|\boldsymbol{x} \otimes \boldsymbol{y}\|_F$$
$$= 2\|\widehat{\boldsymbol{M}} - \boldsymbol{M}\|_F \|\boldsymbol{x}\| \|\boldsymbol{y}\|$$

Let $\boldsymbol{M}_{\text{truncated}}$ be the truncated Tucker approximation of $\boldsymbol{M}$ with multilinear ranks $(R_T, R_T, R_T)$. A priori approximation error bound [32] gives

$$\left\|\boldsymbol{M} - \boldsymbol{M}_{\text{truncated}}\right\|_F \leq \left(\sum_{n=1}^{3} \sum_{k > R_T} \sigma_k^{(n)2}\right)^{1/2}.$$

CP-Decomposing the Tucker core with a size of $(R_T, R_T, R_T)$ yields CP rank at most $R_T^3$. Therefore, the truncated Tucker tensor $\boldsymbol{M}_{\text{truncated}}$ can be written as a CP tensor $\widehat{\boldsymbol{M}}$ with rank at most $R_T^3$; in other words, there exists a CP tensor $\widehat{\boldsymbol{M}}$ such that $\mathrm{rank}_{\text{CP}}(\widehat{\boldsymbol{M}}) \leq R_T^3$ and

$$\left\|\boldsymbol{M} - \widehat{\boldsymbol{M}}\right\|_F \leq \left\|\boldsymbol{M} - \boldsymbol{M}_{\text{truncated}}\right\|_F \leq \left(\sum_{n=1}^{3} \sum_{k > R_T} \sigma_k^{(n)2}\right)^{1/2}.$$

Inserting this to Eq. (7) yields

$$\varepsilon(\boldsymbol{R}, \boldsymbol{x}, \boldsymbol{y}) \leq 2C^2 \left(\sum_{n=1}^{3} \sum_{k > R_T} \sigma_k^{(n)2}\right)^{1/2}.$$

$\square$

## B.2 Universality of CP-Based Tensor Product

**Theorem B.2 (Universality of CP Decomposition):** *Consider* $\mathrm{SO}(3)$*-representations* $\boldsymbol{x} \in V_1 \cong \mathbb{R}^{d_1}$, $\boldsymbol{y} \in V_2 \cong \mathbb{R}^{d_2}$ *and co-domain* $V_3 \cong \mathbb{R}^{d_3}$. *For any* $\mathrm{SO}(3)$*-equivariant bilinear map* $\Phi$*, there exist* $\boldsymbol{B} \in \mathbb{R}^{d_1 \times R}$, $\boldsymbol{C} \in \mathbb{R}^{d_2 \times R}$, $\boldsymbol{A} \in \mathbb{R}^{d_3 \times R}$ *such that* $\Phi$ *can be written as*

$$\Phi(\boldsymbol{x}, \boldsymbol{y}) = \boldsymbol{A}(\boldsymbol{B}^\top \boldsymbol{x} \odot \boldsymbol{C}^\top \boldsymbol{y}) \in V_3,$$

*with* $R \leq d_1 d_2$.

*Proof.* A bilinear map $\Phi$ is uniquely encoded by a third-order tensor $\boldsymbol{T} \in V_3 \otimes V_1^* \otimes V_2^*$ via $\Phi(\boldsymbol{x}, \boldsymbol{y}) = \boldsymbol{T}(\boldsymbol{x} \otimes \boldsymbol{y})$ and all equivariant tensors $\boldsymbol{T}$ form the subspace $\mathcal{H} = (V_3 \otimes V_1^* \otimes V_2^*)^{\mathrm{SO}(3)} = \mathrm{Hom}_{\mathrm{SO}(3)}(V_1 \otimes V_2, V_3)$. For simplicity, we consider the multiplicity-free case $V_i \cong \bigoplus_{\ell=0}^L H^{(\ell)}$, and the general case replaces the scalars below by linear maps on multiplicity spaces. Decomposing $\mathcal{H}$ into irreps gives

$$\mathcal{H} \cong \bigoplus_{\ell_1, \ell_2, \ell_3} \mathrm{Hom}_{\mathrm{SO}(3)}\big(H^{(\ell_1)} \otimes H^{(\ell_2)}, H^{(\ell_3)}\big).$$

By the Clebsch-Gordan decomposition, $\mathrm{Hom}_{\mathrm{SO}(3)}(H^{(\ell_1)} \otimes H^{(\ell_2)}, H^{(\ell_3)})$ is one-dimensional when $|\ell_1 - \ell_2| \leq \ell_3 \leq \ell_1 + \ell_2$ and zero otherwise. For each path $(\ell_1, \ell_2, \ell_3)$, there is a unique map $C_{\ell_1, \ell_2}^{\ell_3} : H^{(\ell_1)} \times H^{(\ell_2)} \to H^{(\ell_3)}$ with matrix elements the Clebsch-Gordan coefficients $C_{\ell_1, m_1, \ell_2, m_2}^{\ell_3, m_3}$, as presented by Eq. (1). Therefore, any equivariant bilinear map $\Phi$ admits the expansion

$$\Phi = \sum_{\ell_1, \ell_2, \ell_3} \alpha_{\ell_1, \ell_2}^{\ell_3} \, C_{\ell_1, \ell_2}^{\ell_3},$$

for some scalars $\alpha_{\ell_1, \ell_2}^{\ell_3}$. Now we index coordinates of $V_1, V_2, V_3$ by $i = (\ell_1, m_1)$, $j = (\ell_2, m_2)$, and $k = (\ell_3, m_3)$. Let the third-order tensor $\boldsymbol{T}_{kij} = \alpha_{\ell_1, \ell_2}^{\ell_3} C_{\ell_1, m_1, \ell_2, m_2}^{\ell_3, m_3}$ and then $\Phi(\boldsymbol{x}, \boldsymbol{y})_k = \sum_{i,j} \boldsymbol{T}_{kij} \boldsymbol{x}_i \boldsymbol{y}_j$. To write $\Phi$ in the CP decomposition form, we take $R = d_1 d_2$ and index the column dimension by pairs $(i, j)$. Define $\boldsymbol{B} \in \mathbb{R}^{d_1 \times R}$, $\boldsymbol{C} \in \mathbb{R}^{d_2 \times R}$, and $\boldsymbol{A} \in \mathbb{R}^{d_3 \times R}$ by

$$\boldsymbol{B}_{i', (i,j)} = \mathbf{1}[i' = i], \qquad \boldsymbol{C}_{j', (i,j)} = \mathbf{1}[j' = j], \qquad \boldsymbol{A}_{k, (i,j)} = \boldsymbol{T}_{kij}.$$

with $\mathbf{1}[\cdot]$ the indicator function. Then $(\boldsymbol{B}^\top \boldsymbol{x})_{(i,j)} = \boldsymbol{x}_i$ and $(\boldsymbol{C}^\top \boldsymbol{y})_{(i,j)} = \boldsymbol{y}_j$, $(\boldsymbol{B}^\top \boldsymbol{x} \odot \boldsymbol{C}^\top \boldsymbol{y})_{(i,j)} = \boldsymbol{x}_i \boldsymbol{y}_j$, and therefore

$$\big(\boldsymbol{A}(\boldsymbol{B}^\top \boldsymbol{x} \odot \boldsymbol{C}^\top \boldsymbol{y})\big)_k = \sum_{i,j} \boldsymbol{T}_{kij} \boldsymbol{x}_i \boldsymbol{y}_j = \Phi(\boldsymbol{x}, \boldsymbol{y})_k.$$

In practice, we approximate $\boldsymbol{T}$ by a lower-rank CP decomposition with computed $\boldsymbol{A}, \boldsymbol{B}, \boldsymbol{C}$, and $R \ll d_1 d_2 \propto (L+1)^4$, corresponding to Fig. 2. $\qquad\square$

# C Model and Training Configurations

**PubChemQCR baseline model configuration.** Table 6 summarizes the configurations for all other baseline models. SchNet [18] and PaiNN [4] are used as implemented in the FAIRChem repository v1. FAENet follows their OC20 release with an $O(3)$ stochastic frame and the "simple" message-passing variant. For MACE [8], we include the real-agnostic residual interaction block. For PACE [33], we retain its interaction block and set the edge-booster dimension to 256. For NequIP [9], MACE, Allegro [35], SevenNet [34], and PACE, we adapt the official repositories to PyTorch Geometric and use a Bessel basis with polynomial cutoff smoothing, keeping all numerical settings at their defaults except the irrep settings. For these models, we set identical irrep-channel dimensions according to Table 6 across irrep blocks. For Equiformer [7], each graph-transformer layer uses 4 attention heads, and the irreps embedding comprises 128 scalars and 64 vectors. All tensor-product-based methods, including NequIP, MACE, Allegro, SevenNet, PACE, and Equiformer, only retain even-parity irreps and use $L_{max} = 2$ except Equiformer for fast training. In addition, all baseline models are trained with gradient-based force prediction except Equiformer for fast training.

Table 6: Configurations including layer counts, hidden (maximum irrep-channel) dimensions, and batch sizes of baseline models including SchNet [18], PaiNN [4], MACE [8], Equiformer [7], PACE [33], FAENet [10], NequIP [9], and Allegro [35] for PubChemQCR experiments.

| MODEL | LAYERS | HIDDEN DIMENSION | BATCH SIZE |
|---|---|---|---|
| SCHNET | 4 | 128 | 128 |
| PAINN | 4 | 128 | 32 |
| FAENET | 4 | 128 | 64 |
| NEQUIP | 5 | 64 | 16 |
| SEVENNET | 5 | 128 | 16 |
| MACE | 2 | 128 | 8 |
| PACE | 2 | 128 | 8 |
| ALLEGRO | 2 | 128 | 8 |
| EQUIFORMER | 4 | 128 | 32 |

**TDN model configuration.** For PubChemQCR and PubChemQCR-S datasets, TDN employs six graph-transformer layers with MLP attention, an irreps-channel dimension of 256, a maximum angular degree $L = 1$, and a graph-transformer layer for the force output head. For the OC20 IS2RE-DIRECT task, TDN adopts the same model configuration and builds the radius graph on the fly with a cutoff of 5.0 and 500 neighbors. For the OC22 IS2RE task, TDN also uses a similar configuration with six graph-transformer layers, a cutoff of 12.0 with 20 neighbors, and an additional degree-9 BOO feature [25] adding to the initial node embedding.

# D Additional Ablation Studies

**Time ablation of path-weight sharing and CP decomposition.** Table 7 presents the time ablation study of path-weight sharing and CP decomposition over the TDN model. As shown in the table, CP decomposition significantly increases the GPU and CPU throughput while path-weight sharing mechanism substantially reduces the parameter count of the model. Note that because TDN removes the depth-wise tensor-product operator, a TDN without CP decomposition and without path-weight sharing in the tensor-product and equivariant-linear layers is not identical to the vanilla Equiformer. All experiments are run on a single NVIDIA A100-80GB GPU and Xeon Gold 6258R processor with a batch size of 128. All experiments are conducted under identical irrep configurations across varying $L$ with 256 irrep-channel dimension, six graph transformer layers, and 8 attention heads of each layer.

Table 7: GPU Throughput and parameter count for Equiformer and TDN variants with or without CP decomposition (CP), path-weight sharing tensor product (PS), path-weight sharing equivariant linear layer (PS-Linear) across maximum degree $L$. Values in parentheses indicate CPU throughput.

| $L$ | MODEL / VARIANT | THROUGHPUT (samples/sec) | PARAMS |
|---|---|---|---|
| | EQUIFORMER | 311.7 (7.5) | 12.1 |
| | TDN | 770.8 (20.2) | 4.5 |
| 1 | TDN *w/o CP* | 328.1 | 4.5 |
| | TDN *w/o CP + PS* | 320.6 | 5.0 |
| | TDN *w/o CP + PS + PS-Linear* | 317.8 | 9.1 |
| | EQUIFORMER | 71.9 (2.4) | 27.9 |
| | TDN | 312.4 (8.7) | 4.5 |
| 2 | TDN *w/o CP* | 83.7 | 4.5 |
| | TDN *w/o CP + PS* | 82.5 | 6.3 |
| | TDN *w/o CP + PS + PS-Linear* | 82.1 | 14.6 |
| | EQUIFORMER | 26.1 (0.6) | 54.7 |
| | TDN | 220.4 (5.8) | 4.5 |
| 3 | TDN *w/o CP* | 26.2 | 4.5 |
| | TDN *w/o CP + PS* | 25.6 | 8.9 |
| | TDN *w/o CP + PS + PS-Linear* | 25.6 | 21.3 |

**Ablation study of CP decomposition rank.** To further demonstrate the practical implication of CP-decomposition-based tensor product and the adopted scheduler, we conduct the ablation study of ranks over $n$-body system dataset [44]. In Table 8 of the n-body system experiment, the TDN is trained with a maximum angular degree $L = 2$, three interaction blocks, and a hidden dimension of 72 in 5000 epochs over a single NVIDIA RTX 2080Ti-11GB GPU. As shown in the table, our adopted schedule uses much smaller ranks yet matches the accuracy obtained with the largest ranks, corresponding to the results of rank schedule selection in. Note that the largest rank is deduced from the upper bound in Section 3.3.

Table 8: TDN Rank-sweep table of the $n$-body system experiment. The last line is TDN without CP decomposition (CP) and path-weight sharing tensor product (PS).

| $R$ | $n$-BODY TESTING MSE | EQUIVARIANCE ERROR |
|---|---|---|
| 10 | 0.0081 | 0.60 |
| 15 | 0.0064 | 0.42 |
| 20 | 0.0051 | 0.21 |
| 28 (OUR SCHEDULER) | 0.0040 | 0.02 |
| 81 (HIGHEST RANK) | 0.0039 | < 0.01 |
| TDN *w/o CP + PS* | 0.0038 | - |

**Ablation of CP-Decomposition-Based Tensor Product for TDN and NequIP.** Under the settings described in Section 4.6, Table 9 shows that for TDN trained for 60 epochs, enabling both mechanisms yields slightly higher validation energy MAE and force RMSE while reducing training time by roughly 78%, suggesting only a minor effect on predictive accuracy. We also evaluate NequIP [9] trained for 100 epochs. Adding the path-weight sharing tensor product and CP decomposition preserves downstream accuracy while cutting computational cost nearly 74%. Note that accuracy metrics are reported from abbreviated training runs intended only to assess relative trends; the fully tuned and converged results are reported in Table 1.

Table 9: Ablation study of the path-weight sharing tensor product (PS) and CP-decomposition-based tensor product (CP). Results are shown for TDN and NequIP [9] on PubChemQCR-S dataset.

| | VALIDATION | | |
|---|---|---|---|
| MODEL | ENERGY MAE (meV/atom) ↓ | FORCE RMSE (meV/Å) ↓ | TRAINING TIME (min/epoch) ↓ |
| TDN *w/o CP + PS* | 12.74 | 92.74 | 19.0 |
| TDN | 12.95 | 96.53 | **4.2** ($\times$ 0.22) |
| NEQUIP | 11.22 | 90.09 | 7.5 |
| NEQUIP + *CP + PS* | 11.15 | 92.49 | **2.0** ($\times$ 0.26) |

