# OpenReview forum: "Tensor Decomposition Networks for Fast Machine Learning Interatomic Potential Computations"
_NeurIPS.cc/2025/Conference — NeurIPS 2025 poster_

### Official Review · Reviewer_S6y1 · 2025-06-16

**Clarity:** 2
**Significance:** 3
**Originality:** 2
**Rating:** 4
**Confidence:** 3

**Summary:**

In this paper, the authors propose the CANDECOMP/PARAFAC (CP) low-rank tensor decomposition to approximate the tensor involved in the Clebsch-Gordan (CG) tensor product, commonly used in dominant machine learning force field (MLFF) models. These models, based on equivariant graph neural networks, incorporate rotational symmetry — a fundamental property of molecular and material systems. In addition to the CP approximation, a path-weight sharing strategy is proposed that further reduces the number of learnable parameters, leading to a total computational complexity reduction of the CG tensor product from O(L6) to O(L4). The impact of the proposed method is evaluated on the computation time, approximation accuracy, and equivariance preservation using two benchmarks: the PubChemQC-Traj dataset, which contains millions of molecular dynamics snapshots, and the Open Catalyst Project (OCP), a large-scale dataset of DFT relaxations for adsorbate–catalyst systems.

**Questions:**

1-	The method for selecting the optimal rank R is not clearly explained. Is R tuned via cross-validation?

2-	Which dataset Figure 3 refers to?

3-	How do you define the “wall-clock runtime” that you evaluated in the experiments? Is it related to computation time during training, testing or both?

**Ethical Concerns:**

["NO or VERY MINOR ethics concerns only"]

**Final Justification:**

After reading the author's responses and other reviews, I decided to keep my rating. However, I thank the authors for clarifying my misunderstanding of Fig. 3 and the term "wall-clock runtime" used in the paper.

**Limitations:**

yes

**Paper Formatting Concerns:**

No concerns

**Quality:**

3

**Strengths And Weaknesses:**

Strengths

•	The results presented in this paper are relevant to the development of force field models that incorporate rotational invariance, an important property for molecular and material systems.

•	The paper provides theoretical bounds on both approximation and equivariance errors, derived from the singular values of the matrix representations of the approximated tensor MM.

Weaknesses

•	The low-rank structure of the tensor M deserves further discussion. While the paper relates the rank R to the maximum angular degree L, a theoretical justification or empirical analysis of the inherent low-rank nature of M is lacking.

•	The method for selecting the optimal rank R is not clearly explained. Is R tuned via cross-validation? Clarifying whether this is a learned or predefined hyperparameter is needed.

•	The theoretical analysis focuses on the forward computation of the tensor product. However, during training, the gradients with respect to the CP decomposition parameters must also be computed. Therefore, an analysis of the computational complexity of gradient backpropagation should be included.

•	Figure 3: It is unclear which dataset this figure refers to. This should be explicitly stated in the caption or text.

•	The evaluation of computation time is described in terms of “wall-clock runtime” but it is ambiguous whether this refers to training time, inference time, or both. A more precise description of the measurement setup is needed.

•	In Line 285, a reference to a table is made, but the table number is missing.

---

> ### Author Rebuttal · Authors · 2025-07-31
>
> We appreciate that Reviewer S6y1 finds our method significant.
>
>
> > The low-rank structure of the tensor M deserves further discussion. While the paper relates the rank R to the maximum angular degree L, a theoretical justification or empirical analysis of the inherent low-rank nature of M is lacking.
>
> Our work mainly approximates the Clebsch Gordan (CG) tensor product via CP decomposition. The CG coefficient tensor is intrinsically low-rank because of the sparsity that most index triples $(\ell_1, \ell_2, \ell_3)$ in the CG tensor violate the selection rules $ |\ell_1 - \ell_2| \le \ell_3 \le \ell_1 + \ell_2 $ and $m_3 = m_1 + m_2$; their entries are therefore identically zero.
>
> Empirical evidence supports this view. By the upper bound of the tensor rank [1], the maximum rank of a CG tensor is $(L+1)^4 \propto O(L^4)$ while we successfully employ only an $O(L^2)$ rank schedule to approximate the CG tensor. Below we present the ablation study of ranks over *n*-body system dataset [2], where the TDN is trained with a maximum angular degree $L=2$, 4 interaction blocks, and a hidden dimension of 64 in 5000 epochs. In the table, our adopted schedule uses much smaller ranks yet matches the accuracy obtained with the largest ranks, corresponding to the results of rank schedule selection in Section 4.1 and showing the low rank nature of the CG tensor.
>
> | R                          | *n*-Body Testing MSE | Equivariance Error |
> | -------------------------- | ----------------------- | ------------------ |
> | 10                         | 0.00910                 | 0.60               |
> | 15                         | 0.00825                 | 0.42               |
> | 20                         | 0.00648                 | 0.21               |
> | **28 (Our schdule)**      | 0.00438                 | 0.02               |
> | 81 (Highest possible rank) | 0.00437                 | <1e-3              |
>
>
>
> > The method for selecting the optimal rank R is not clearly explained. Is R tuned via cross-validation? Clarifying whether this is a learned or predefined hyperparameter is needed.
>
> > The method for selecting the optimal rank R is not clearly explained. Is R tuned via cross-validation?
>
> Rank $R$ is a **predefined hyperparameter**, not a learned quantity. Because computing the exact CP rank of the CG tensor is NP-hard, we instead explore several simple growth rules before network training (Section 4.1) to guide the selection of rank $R$. Our adopted schedule gives a relatively small SO(3)-equivariance errors with low computational cost as demonstrated in Figure 3, and rank-sweep ablations of $L=2$ in the above table confirm that this schedule suffices for downstream tasks.
>
> Although this heuristic meets the needs of our current force‑field tasks, a data‑driven approach, such as learning a rank predictor or applying Bayesian optimization over ranks, could generalize the method and further reduce redundancy; we identify this as an important direction for future work.
>
>
> > The theoretical analysis focuses on the forward computation of the tensor product. However, during training, the gradients with respect to the CP decomposition parameters must also be computed. Therefore, an analysis of the computational complexity of gradient backpropagation should be included.
>
>
> We present both the **theoretical and empirical time complexity analysis** of the CP decomposition for the tensor product. Let $A\in \mathbb{R}^{d_3\times R}, B\in \mathbb{R}^{d_1\times R}, C\in \mathbb{R}^{d_2\times R}$ and input $x\in \mathbb{R}^{d_1}, y\in \mathbb{R}^{d_2}$. By Equation (3), the forward map is
> $$
> z = A(B^\top x\odot C^\top y ).
> $$
> Given loss gradient $g_z = \frac{\partial \mathcal L}{\partial z}\in \mathbb{R}^{d_3}$, the backward gradient to $x$ is
> $$
> g_x = g_z\frac{\partial{A(B^\top x\odot C^\top y )}}{\partial x} = g_z\frac{\partial A \text{diag}(C^\top y) B^\top x}{\partial x} = g_z A \text{diag}(C^\top y) B^\top = (g_z A \odot C^\top y) B^\top
> $$
> and similarly the backward gradient to $y$ is
> $$
> g_y = (g_z A \odot B^\top x) C^\top.
> $$
> Each expression requires a time complexity of $O(R(d_1+d_2+d_3))$. Therefore, the total time complexity to compute the backward gradient is $O(R(d_1+d_2+d_3))$, identical to the forward pass.
>
> Furthermore, the computation time of both forward and backward is presented in the below table. Each pass is measured with a batch size of 1000 and a multiplicity of 128. The backward step is consistently $\approx 2-3×$ the forward cost, independent of L, confirming that the same time complexity of forward and backward pass.
>
>
> |                                            | Forward Computation Time | Backward Computation Time |
> | ------------------------------------------ | ------- | -------- |
> | L = 1 | 0.42 ms | 1.29 ms  |
> | L = 4 | 0.57 ms | 1.53 ms  |
> | L = 6 | 0.83 ms | 2.24 ms  |
>
>
>
> > Figure 3: It is unclear which dataset this figure refers to. This should be explicitly stated in the caption or text.
>
> > Which dataset Figure 3 refers to?
>
> **Figure 3 is not tied to any dataset**. The curves in Figure 3 report the approximation error and runtime of the CP decomposition for the CG tensor product itself, which only depends on the chosen rank, multiplicities, and maximum angular degree. The low-rank reconstruction can be learned and evaluated without any molecular data. To learn the CP factor matrices in CP decomposition, we minimize the Frobenius norm between the exact and approximate CG tensor product on batches of standard normal inputs; this guards against data bias and is done once before any network training. We will amend the caption to state this explicitly in Figure 3 in the camera‑ready version.
>
>
> > The evaluation of computation time is described in terms of “wall-clock runtime” but it is ambiguous whether this refers to training time, inference time, or both. A more precise description of the measurement setup is needed.
>
> > How do you define the “wall-clock runtime” that you evaluated in the experiments? Is it related to computation time during training, testing or both?
>
> The "wall-clock runtime" refers to the inference time over the PubchemQC-Traj-Small dataset. We will revise line 260 to state this explicitly in the camera‑ready version.
>
>
> > In Line 285, a reference to a table is made, but the table number is missing.
>
> The table reference should be "Table 3". We will revise line 285 and update this in our camera-ready version.
>
>
> [1] Kolda, T. and Bader, B. Tensor decompositions and applications. SIAM review, 51(3):455–500, 2009.
>
>
> [2] Kipf, T., Fetaya, E., Wang, K.-C., Welling, M., and Zemel, R. Neural relational inference for interacting systems. In International conference on machine learning, pp. 26882697. PMLR, 2018.

---

> > ### Comment · Reviewer_S6y1 · 2025-08-05
> > **Acknowledgment of responses**
> >
> > I thank the authors for addressing all my questions and comments

---

> ### Author Response · Authors · 2025-08-06
> **Official Comment by Authors**
>
> Dear Reviewer S6y1,
>
> Thank you again for the positive evaluation and constructive feedback. As we approach the end of the reviewer-author discussion period, we would like to kindly check if there are any new questions or concerns you would like us to address. If our rebuttal has fully addressed your concerns, we would be grateful if you might consider increasing your score to reflect a solid accept.
>
> Best regards,
>
> Authors

---

### Official Review · Reviewer_oXft · 2025-06-18

**Clarity:** 3
**Significance:** 3
**Originality:** 3
**Rating:** 4
**Confidence:** 4

**Summary:**

This manuscript presents an interesting and timely contribution to the development of efficient SO(3)-equivariant neural networks. It addresses a known computational bottleneck in machine learning force fields: the expensive Clebsch–Gordan (CG) tensor product. The key innovation lies in approximating spherical tensors via low-rank decompositions; specifically, using CANDECOMP/PARAFAC (CP) decomposition to express the CG tensor as a sum of rank-1 tensors.

**Questions:**

Missing some important citation:

Brandon Anderson, Truong-Son Hy and Risi Kondor, Cormorant: Covariant molecular neural networks, NeurIPS 2019.

https://dl.acm.org/doi/10.5555/3454287.3455589

**Ethical Concerns:**

["NO or VERY MINOR ethics concerns only"]

**Final Justification:**

I have read the authors' rebuttal as well as other reviewers' comments. The authors have added more experimental results. I decided to keep my original rating as borderline accept.

**Limitations:**

- Approximate Equivariance: The method only maintains approximate SO(3)-equivariance. While the authors provide theoretical bounds on the equivariance error, the practical implications of this approximation, especially in sensitive downstream tasks, are not deeply analyzed.

- CP Rank Selection is Manual: The required CP rank R is treated as a fixed hyperparameter. Since determining the optimal CP rank is NP-hard, this introduces potential inefficiency. A more adaptive or data-driven approach to rank selection would improve scalability and generality.

- Limited Scope of Decomposition: The proposed CP decomposition is demonstrated only for the Clebsch–Gordan tensor product. It's unclear how well the approach generalizes to other types of equivariant operations (for example, permutation-equivariance: please see "Predicting molecular properties with covariant compositional networks" at Journal of Chemical Physics https://pubs.aip.org/aip/jcp/article/148/24/241745/964420/Predicting-molecular-properties-with-covariant).

- The evaluation is mostly limited to molecular force field tasks (energy and force prediction). It would strengthen the paper to show broader utility in other domains where SO(3)-equivariance is important (e.g., protein modeling, 3D object recognition).

- No Ablation on Universality vs. Accuracy Trade-off: Although the paper proves universality theoretically, it lacks an ablation showing how the approximation degrades or improves as R increases. Practical guidance for choosing R would be valuable.

- The path-weight sharing mechanism drastically reduces parameter count by using a single weight matrix across all paths. While this helps with efficiency, it may limit the expressiveness or fine-grained control across angular degrees, especially at high L.

**Quality:**

3

**Strengths And Weaknesses:**

Strengths:
- Novelty: The idea of replacing the CG product with a CP decomposition is elegant and reduces computational complexity from O(L^6) to O(L^4), a substantial improvement.

- Theory: The authors provide theoretical guarantees, including equivariance error bounds and a universality theorem for their approximation.

- Efficiency: They combine CP decomposition with a path-weight sharing scheme to reduce parameters significantly (from O(c^2 L^3) to O(c^2)).

- Empirical results: Strong performance is demonstrated on PubChemQC-Traj, OC20, and OC22, with impressive speed-ups and competitive accuracy.

Weaknesses:
- While the approximation method is justified theoretically, the required CP rank R is treated as a hyperparameter. Estimating or learning optimal ranks adaptively could improve generality.

- The approximation is "approximately equivariant". The trade-off between error and efficiency for approximate equivariance has been well-explored in the literature. I think this work could benefit from more discussion in real-world generalization contexts.

---

> ### Author Rebuttal · Authors · 2025-07-31
>
> We appreciate that Reviewer oXft finds our method significant.
>
> > While the approximation method is justified theoretically, the required CP rank R is treated as a hyperparameter. Estimating or learning optimal ranks adaptively could improve generality.
>
> > CP Rank Selection is Manual: The required CP rank R is treated as a fixed hyperparameter. Since determining the optimal CP rank is NP-hard, this introduces potential inefficiency. A more adaptive or data-driven approach to rank selection would improve scalability and generality.
>
> Selecting the CP rank $R$ is itself an NP‑hard problem as mentioned in the limitation section, so in practice **the rank schedules are empirical heuristics**. The adopted one gives a relatively small SO(3)-equivariance error with low computational cost across every schedule in Figure 3 and Section 4.1, and ablation studies in the below table of NequIP confirm that this schedule suffices for downstream tasks while cutting time cost substantially. Note that the NequIP is trained with a maximum angular degree $L=2$, 4 interaction blocks, and a multiplicity of 64 for 100 epochs with our proposed schedule.
>
> |  NequIP                                        | Validation Energy MAE (eV) | Validation Force RMSE (eV/Å) | Training Time  (min/epoch) | Inference Time (min/epoch) |
> | ---------------------------------------- | -------------------------- | ----------------------------- | -------------------------- | -------------------------- |
> | Plain                                    | 0.01122                    | 0.09009                       | 7.5                        | 1.5                        |
> | + Path-weight sharing & CP decomposition | 0.01115                    | 0.09249                       | 2.0                        | 0.5                        |
>
> Although this heuristic meets the needs of our current force‑field tasks, we agree that a data‑driven approach, such as learning a rank predictor or applying Bayesian optimization over ranks, could generalize the method and further reduce redundancy; we identify this as an important direction for future work.
>
>
> > The approximation is "approximately equivariant". The trade-off between error and efficiency for approximate equivariance has been well-explored in the literature. I think this work could benefit from more discussion in real-world generalization contexts.
>
>
> We agree that the value of approximate equivariance ultimately rests on how well the model generalizes. We share the concern and therefore measured our CP-decomposition-based approximation on two large-scale, experiment-derived benchmarks: Open Catalyst Project and PubChemQC-Traj. Both datasets contain millions of chemically diverse geometries and are labelled with high-fidelity DFT energies and forces. We believe the performance on these two datasets is a strong indicator of generalization ability.
>
> To push the generalization study further, we plan to train on OMol [1], a newly released molecular dataset that spans even broader compositional space. These additional experiments may help clarify how the approximation behaves on systems that differ more sharply from the training distribution.
>
> > Missing some important citation: Brandon Anderson, Truong-Son Hy and Risi Kondor, Cormorant: Covariant molecular neural networks, NeurIPS 2019.
>
>
> Thank you for raising this issue. We will update the citation at line 25 and line 72 in the camera-ready version.
>
>
> > Approximate Equivariance: The method only maintains approximate SO(3)-equivariance. While the authors provide theoretical bounds on the equivariance error, the practical implications of this approximation, especially in sensitive downstream tasks, are not deeply analyzed.
>
> > No Ablation on Universality vs. Accuracy Trade-off: Although the paper proves universality theoretically, it lacks an ablation showing how the approximation degrades or improves as R increases. Practical guidance for choosing R would be valuable.
>
>
> Thank you for raising these points. To further demonstrate the practical implication of this approximation and strengthen the contribution, we conduct the **ablation study of ranks over *n*-body system dataset** [2]. In the below rank‑sweep table for the *n*-body system experiment, the TDN is trained with a maximum angular degree $L=2$, 4 interaction blocks, and a hidden dimension of 64 in 5000 epochs. In that table, our adopted schedule uses much smaller ranks yet matches the accuracy obtained with the largest ranks, corresponding to the results of rank schedule selection in Section 4.1. Note that the largest rank is deduced from the upper bound in line 164; see [3].
>
>
> | R                          | *n*-Body Testing MSE | Equivariance Error |
> | -------------------------- | ----------------------- | ------------------ |
> | 10                         | 0.00910                 | 0.60               |
> | 15                         | 0.00825                 | 0.42               |
> | 20                         | 0.00648                 | 0.21               |
> | **28 (Our schdule)**      | 0.00438                 | 0.02               |
> | 81 (Highest possible rank) | 0.00437                 | <1e-3              |
>
>
>
>
> > Limited Scope of Decomposition: The proposed CP decomposition is demonstrated only for the Clebsch–Gordan tensor product. It's unclear how well the approach generalizes to other types of equivariant operations (for example, permutation-equivariance).
>
>
> Tensor product is widely used in SO(3)-equivariant neural networks. Particularly, the **Clebsch-Gordan tensor product is the canonical formulation of an SO(3)-equivariant neural network**. For permutation equivariance, to our knowledge, it is unknown that tensor product will play a role in it. The permutation-equivariant application is usually built by deep sets or graph neural networks, where bilinear operations are not necessarily used. For other types of equivariant operations, as long as they can be written in bilinear forms, our CP decomposition can be applied to approximate the operations.
>
>
> > The evaluation is mostly limited to molecular force field tasks (energy and force prediction). It would strengthen the paper to show broader utility in other domains where SO(3)-equivariance is important (e.g., protein modeling, 3D object recognition).
>
>
> We appreciate this suggestion. Our study deliberately targets machine-learning interatomic potentials since accelerating the Clebsch Gordan map yields a great practical benefit there, and the benchmarks (Open Catalyst Project, PubChemQC-Traj) are well established. Extending the approach to other domains, e.g., protein folding and 3-D object recognition, is indeed an exciting next step, and we leave it as future explorations.
>
>
>
> > The path-weight sharing mechanism drastically reduces parameter count by using a single weight matrix across all paths. While this helps with efficiency, it may limit the expressiveness or fine-grained control across angular degrees, especially at high L.
>
>
> We agree that path-weight sharing will limit the expressiveness of the tensor product. However, **coupling with CP decomposition, the result can match the full Clebsch Gordan (CG) map**. To demonstrate this, we further conduct the ablations of CP decomposition and path-weight sharing for both the ***n*-body experiment dataset** and a **subset of PubchemQC-Traj-Small dataset** in the below table. For the *n*-body experiment, the testing results of TDN are presented in the table, where the TDN is trained with a maximum angular degree $L=2$, 4 interaction blocks, a hidden dimension of 72, and a weight decay of $5\times 10^{-6}$ in 2000 epochs. And for the subset of PubchemQC-Traj-Small dataset, we randomly select one conformer for each trajectory from PubchemQC-Traj-Small to build the training, validation, and testing set. The testing results of TDN over this subset are presented in the table, where the TDN is trained with a maximum angular degree $L=2$, 4 interaction blocks, and a hidden dimension of 64 in 60 epochs. Note that in the table, the *n*-Body experiment appears under the label "*n*-Body," while the subset of the PubChemQC-Traj-Small dataset is labeled simply as "Subset."
>
>
> |                                          | *n*-Body MSE | Subset Testing Energy MAE (eV) | Subset Testing Force RMSE (eV/Å) | Subset Training Time Cost (min/epoch) |
> | ---------------------------------------- | ------------ | ------------------------------ | -------------------------------- | ------------------------------------- |
> | Plain                                    | 0.00414      | 0.01274                        | 0.09274                          | 19.0                                  |
> | + Path-weight sharing                    | 0.00439      | 0.01503                        | 0.09321                          | 9.8                                   |
> | + Path-weight sharing & CP decomposition | 0.00429      | 0.01295                        | 0.09653                          | 4.2                                   |
>
> As shown in the above table, introducing path-weight sharing alone will indeed incur a small accuracy drop. However, introducing both path-weight sharing and CP decomposition produces comparable results to the full CG tensor product while significantly boosting the efficiency. We believe this is because the Clebsch Gordan map is replaced by a low-rank surrogate with rank-1 factors, which act as an implicit regularizer as they filter out high-variance directions that weight sharing alone cannot suppress. We will include this table in the appendix of the camera‑ready version.
>
> [1] Levine, D. S., et al. The open molecules 2025 (omol25) dataset, evaluations, and models.
>
> [2] Kipf, T., et al. Neural relational inference for interacting systems. In International conference on machine learning, pp. 26882697. PMLR, 2018.
>
> [3] Kolda, T. and Bader, B. Tensor decompositions and applications. SIAM review, 51(3):455–500, 2009.

---

> > ### Comment · Area_Chair_bR37 · 2025-08-05
> >
> > Dear Reviewer,
> >
> > As the author-reviewer discussion period is drawing to a close, we kindly ask that you respond to the authors' rebuttal.
> >
> > Thank you for your contributions.
> >
> > Best regards,
> > Your AC

---

> ### Author Response · Authors · 2025-08-06
> **Official Comment by Authors**
>
> Dear Reviewer oXft,
>
> Thank you again for the positive evaluation and constructive feedback. As we approach the end of the reviewer-author discussion period, we would like to kindly check if there are any new questions or concerns you would like us to address. If our rebuttal has fully addressed your concerns, we would be grateful if you might consider increasing your score to reflect a solid accept.
>
> Best regards,
>
> Authors

---

### Official Review · Reviewer_cmnz · 2025-07-02

**Clarity:** 3
**Significance:** 3
**Originality:** 3
**Rating:** 5
**Confidence:** 3

**Summary:**

The work proposes to use tensor decomposition, namely the CANDECOMP/PARAFAC (CP) decomposition, to accelerate the Clebsch-Gordan (CG) tensor product machine learning force field (MLFF) models. Moreover, to achieve further reduction, the authors propose a weight-sharing scheme.

**Questions:**

- Why is EQUIFORMER selected as the main method for acceleration? It would be nice to see if the CPD applicability and performance are not only for EQUIFORMER but also for other tensor-product-based methods.
- Is the GPU runtime reported in Table 2? Can the authors show the CPU runtime as well?
- Would the proposed method benefit from other tensor network (decomposition) structures? What is the author's opinion?

**Ethical Concerns:**

["NO or VERY MINOR ethics concerns only"]

**Final Justification:**

I will raise my original score, as this work is highly significant for both AI and computational material science communities.

**Limitations:**

Yes, the limitations are discussed in the Appendix.

**Paper Formatting Concerns:**

It would be nice to add relative numbers on parameter counts and processing speed directly to Table 2.

**Quality:**

3

**Strengths And Weaknesses:**

**Strengths**
- The addressed problem of compression and acceleration of MLFF models is under-explored in the community but is still very important.
- An error bound analysis is provided.
- A rank selection scheme suitable for the method is presented.

**Weaknesses**
Please address the questions section in the relevant section.

---

> ### Author Rebuttal · Authors · 2025-07-31
>
> We appreciate that Reviewer cmnz finds our method significant.
>
> > Why is EQUIFORMER selected as the main method for acceleration? It would be nice to see if the CPD applicability and performance are not only for EQUIFORMER but also for other tensor-product-based methods.
>
> We focused on Equiformer [1] because Equiformer uses the fully-connected Clebsch Gordan (CG) tensor product as the basic equivariant operation. This is the canonical formulation of an SO(3)‑equivariant network and is significant for the machine learning force field prediction task [1-3]. Also, the fully-connected CG tensor product is a cost‑dominating kernel so the acceleration is easy to measure. Furthermore, Equiformer outperforms many other baselines with fully-connected CG tensor products in several molecular tasks [1]. Considering all the above, Equiformer provides a standard and widely recognized baseline.
>
> **The CP‑decomposition method itself is not tied to Equiformer**. Any architecture whose core step is a bilinear operation, e.g., Tensor Field Networks (TFN) [2], NequIP [3], or other e3nn “TensorProduct” blocks, can be approximated in the same way. We further provide the ablation study over the NequIP architecture [3] with and without our techniques in the below table. The employed dataset is the subset of PubchemQC-Traj-Small dataset, where we randomly select one conformer for each trajectory from PubchemQC-Traj-Small to build the training, validation, and testing set. And the NequIP is trained with a maximum angular degree $L=2$, 4 interaction blocks, and a multiplicity of 64 for 100 epochs. As shown in the below table, the energy and force validation results are similar, but the training and inference time are accelerated by 2 times with our techniques.
>
> |  NequIP                                        | Validation Energy MAE (eV) | Validation Force RMSE (eV/Å) | Training Time  (min/epoch) | Inference Time (min/epoch) |
> | ---------------------------------------- | -------------------------- | ----------------------------- | -------------------------- | -------------------------- |
> | Plain                                    | 0.01122                    | 0.09009                       | 7.5                        | 1.5                        |
> | + Path-weight sharing & CP decomposition | 0.01115                    | 0.09249                       | 2.0                        | 0.5                        |
>
>
>
> > Is the GPU runtime reported in Table 2? Can the authors show the CPU runtime as well?
>
>
> The time reported in Table 2 is the inference time over a single A100-80GB GPU, and the runtime includes both GPU and CPU overhead. The pure CPU implementation runtime over AMD EPYC 9684X 96-Core Processor is detailed in the below table.
>
> |                  | Throughput (samples/sec) |
> | ---------------- | ------------------------ |
> | Equiformer (L=1) | 7.3                      |
> | TDN (L=1)        | 17.0                     |
> | Equiformer (L=2) | 1.8                      |
> | TDN (L=2)        | 7.9                      |
> | Equiformer (L=3) | 0.6                      |
> | TDN (L=3)        | 4.7                      |
>
>
> We will include this CPU table in the appendix of the camera‑ready version.
>
>
> > Would the proposed method benefit from other tensor network (decomposition) structures? What is the author's opinion?
>
>
> The current study uses the CP decomposition because it is the simplest: it requires only rank‑1 components, is easy to fit once, and introduces no new hyperparameters beyond the rank schedule. Additionally, the error bound and the uniqueness of CP decomposition are well studied in [4]. Other tensor decomposition methods are promising as well, such as **Tucker decomposition** [4] and **tensor‑train decomposition** [5]. These alternatives offer a different trade‑off between parameter count, arithmetic complexity, and the cache locality. We view a systematic comparison of these alternatives, and their effect on training stability and accuracy, as valuable future work.
>
>
> > It would be nice to add relative numbers on parameter counts and processing speed directly to Table 2.
>
> We present the updated Table 2 below, where we include the multiplicative ratio with respect to the Equiformer baseline with the same $L$. We will include this table in the camera‑ready version.
>
>
> |                  | Throughput (samples/sec) | Parameter       |
> | ---------------- | ------------------------ | --------------- |
> | Equiformer (L=1) | 470.9                    | 8.86 M          |
> | TDN (L=1)        | 583.1 (x 1.24)           | 4.42 M (x 0.50) |
> | Equiformer (L=2) | 122.5                    | 18.19 M         |
> | TDN (L=2)        | 199.7 (x 1.63)           | 4.42 M (x 0.24) |
> | Equiformer (L=3) | 48.4                     | 33.43 M         |
> | TDN (L=3)        | 142.9 (x 2.95)           | 4.42 M (x 0.13) |
>
>
>
> [1] Liao, Y. and Smidt, T. Equiformer: Equivariant graph attention transformer for 3d atomistic graphs. In The Eleventh International Conference on Learning Representations, 2022.
>
> [2] Thomas, N., Smidt, T., Kearnes, S., Yang, L., Li, L., Kohlhoff, K., and Riley, P. Tensor field networks: Rotation-and translation-equivariant neural networks for 3d point clouds. arXiv preprint arXiv:1802.08219, 2018.
>
> [3] Batzner, S., Musaelian, A., Sun, L., Geiger, M., Mailoa, J. P., Kornbluth, M., Molinari, N., Smidt, T. E., and Kozinsky, B. "E(3)-equivariant graph neural networks for data-efficient and accurate interatomic potentials." Nature communications 13, no. 1 (2022): 2453
>
> [4] Kolda, T. and Bader, B. Tensor decompositions and applications. SIAM review, 51(3):455–500, 2009.
>
> [5] Oseledets, I. V. Tensor-train decomposition. SIAM Journal on Scientific Computing, 33(5), 2295-2317.

---

> > ### Comment · Reviewer_cmnz · 2025-08-06
> >
> > Thank you for addressing the comments.

---

> ### Author Response · Authors · 2025-08-06
> **Official Comment by Authors**
>
> Dear Reviewer cmnz,
>
> Thank you again for the positive evaluation and constructive feedback. As we approach the end of the reviewer-author discussion period, we would like to kindly check if there are any new questions or concerns you would like us to address. If our rebuttal has fully addressed your concerns, we would be grateful if you might consider increasing your score to reflect a solid accept.
>
> Best regards,
>
> Authors

---

> > ### Comment · Reviewer_cmnz · 2025-08-08
> >
> > Thank you for your work. I will increase my original score.

---

### Official Review · Reviewer_7Mvd · 2025-07-03

**Clarity:** 3
**Significance:** 3
**Originality:** 2
**Rating:** 4
**Confidence:** 3

**Summary:**

This paper introduces tensor decomposition networks (TDNs) as a class of approximately equivariant networks whose CG tensor products are replaced by low-rank tensor decompositions, such as the CANDECOMP/PARAFAC (CP) decomposition. With the CP decomposition, this paper proves (i) a uniform bound on the induced error of SO(3)-equivariance, and (ii) the universality of approximating any equivariant bilinear map. To further reduce the number of parameters, this paper proposes path-weight sharing that ties all multiplicity-space weights across the O(L^3) CG paths into a single path without compromising equivariance, where L is the maximum angular degree. The resulting layer acts as a plug-and-play replacement for tensor products in existing networks, and the computational complexity of tensor products is reduced
from O(L^6) to O(L^4). Experiment results show that TDNs achieve competitive performance with dramatic speedup in computations.

**Questions:**

Please see the weakness above.

**Ethical Concerns:**

["NO or VERY MINOR ethics concerns only"]

**Final Justification:**

My questions have been addressed. I keep my score as it was.

**Limitations:**

yes

**Paper Formatting Concerns:**

no concerns on paper formatting

**Quality:**

3

**Strengths And Weaknesses:**

Strengths:

- Introducing tensor decomposition in MLFF and prove its effectiveness is significant for the field.

- This paper provides solid theoretical guarantees (error bounds, universality) for the proposed CP-based tensor approximations.

- Strong empirical results demonstrating TDN's performance improvements on multiple datasets over multiple models.

Weaknesses:

- In line 233, what is the motivation of the proposed rank schedules?

- Lack of ablation study on CP decomposition and path-weight sharing.

- While CP decomposition improves computational efficiency, it also introduces abstraction: the learned rank-1 components may lack clear physical or geometric interpretations.

---

> ### Author Rebuttal · Authors · 2025-07-31
>
> We appreciate that Reviewer 7Mvd finds our method significant. To further strengthen our contribution, we add *n*-body experiments [1,2] as an additional benchmark.
>
> > In line 233, what is the motivation of the proposed rank schedules?
>
> The rank schedules are empirical heuristics and are corresponded to different complexity growth of the CP decomposition. The adopted one gives a relatively small SO(3)-equivariance error with low computational cost across every schedule in Figure 3 and Section 4.1. Further ablations clarify that this schedule gives a good trade-off between efficiency and approximation error:
>
> - **Ablations of ranks at $L=2$** show that using this schedule leaves downstream metrics with negligible impact. In the below rank‑sweep table of the *n*-body system experiment, the TDN is trained with a maximum angular degree $L=2$, 4 interaction blocks, and a hidden dimension of 64 in 5000 epochs. In that table, our schedule uses much smaller ranks yet matches the accuracy obtained with the largest rank to approximate the Clebsch-Gordan tensor product. Note that the largest rank is deduced from the upper bound in line 164; see [3].
>
> - Likewise, in the **ablations of CP-decomposition and path‑weight‑sharing** (in the next reply), turning CP decomposition on and off under our schedule has negligible effect on final performance but markedly improves efficiency.
>
> | R                          | *n*-Body Testing MSE | Equivariance Error |
> | -------------------------- | ----------------------- | ------------------ |
> | 10                         | 0.00910                 | 0.60               |
> | 15                         | 0.00825                 | 0.42               |
> | 20                         | 0.00648                 | 0.21               |
> | **28 (Our schdule)**      | 0.00438                 | 0.02               |
> | 81 (Highest possible rank) | 0.00437                 | <1e-3              |
>
>
> > Lack of ablation study on CP decomposition and path-weight sharing.
>
> Below we present the ablation study of CP decomposition and path-weight sharing for both the ***n*-body experiment dataset** and the **subset of PubchemQC-Traj-Small dataset**. For the *n*-body experiment, the testing results of TDN are presented in the table, where the TDN is trained with a maximum angular degree $L=2$, 4 interaction blocks, a hidden dimension of 72, and a weight decay of $5\times 10^{-6}$ in 2000 epochs. And for the subset of PubchemQC-Traj-Small dataset, we randomly select one conformer for each trajectory from PubchemQC-Traj-Small to build the training, validation, and testing set. The testing results of TDN over this subset are presented in the table, where the TDN is trained with a maximum angular degree $L=2$, 4 interaction blocks, and a hidden dimension of 64 in 60 epochs. Note that in the table, the *n*-Body experiment appears under the label "*n*-Body," while the subset of the PubChemQC-Traj-Small dataset is labeled simply as "Subset."
>
> |                                          | *n*-Body MSE | Subset Testing Energy MAE (eV) | Subset Testing Force RMSE (eV/Å) | Subset Training Time Cost (min/epoch) |
> | ---------------------------------------- | ------------ | ------------------------------ | -------------------------------- | ------------------------------------- |
> | Plain                                    | 0.00414      | 0.01274                        | 0.09274                          | 19.0                                  |
> | + Path-weight sharing                    | 0.00439      | 0.01503                        | 0.09321                          | 9.8                                   |
> | + Path-weight sharing & CP decomposition | 0.00429      | 0.01295                        | 0.09653                          | 4.2                                   |
>
> As shown in the above table, introducing both path-weight sharing and CP decomposition does not affect the performance a lot while significantly boosting the efficiency, although we notice that introducing path-weight sharing alone will hurt the performance a bit. We believe this is because the Clebsch Gordan map is replaced by the low-rank surrogate with rank-1 factors, which act as an implicit regularizer as they filter out high-variance directions that weight sharing alone cannot suppress. We will include this table in the appendix of the camera‑ready version.
>
> > While CP decomposition improves computational efficiency, it also introduces abstraction: the learned rank-1 components may lack clear physical or geometric interpretations.
>
> Thank you for raising this point. Although the rank‑1 components from the CP decomposition are indeed harder to interpret physically, they do not strip away information contained in the Clebsch Gordan tensor if the rank is sufficiently large, as shown in the empirical ablation study above and Theorem 3.2.
>
> Conceptually, Clebsch Gordan coefficients are coordinates of the SO(3) tensor product in the standard coupled-angular-momentum basis. Switching to a CP basis simply means **re-expressing the same sparse linear map in a different compact rank‑1 factor basis**; the transformation is analogous to choosing a new set of axes in linear algebra, and information is re‑packaged.
>
>
> [1] Kipf, T., Fetaya, E., Wang, K.-C., Welling, M., and Zemel, R. Neural relational inference for interacting systems. In International conference on machine learning, pp. 26882697. PMLR, 2018.
>
> [2] Satorras, V. G., Hoogeboom, E., and Welling, M. E(n) equivariant graph neural networks. In International conference on machine learning, pp. 9323–9332. PMLR, 2021.
>
> [3] Kolda, T. and Bader, B. Tensor decompositions and applications. SIAM review, 51(3):455–500, 2009.

---

> > ### Comment · Area_Chair_bR37 · 2025-08-05
> >
> > Dear Reviewer,
> >
> > As the author-reviewer discussion period is drawing to a close, we kindly ask that you respond to the authors' rebuttal.
> >
> > Thank you for your contributions.
> >
> > Best regards,
> > Your AC

---

> > ### Comment · Reviewer_7Mvd · 2025-08-06
> >
> > Thank you for thoroughly addressing my questions.

---

> ### Author Response · Authors · 2025-08-06
> **Official Comment by Authors**
>
> Dear Reviewer 7Mvd,
>
> Thank you again for the positive evaluation and constructive feedback. As we approach the end of the reviewer-author discussion period, we would like to kindly check if there are any new questions or concerns you would like us to address. If our rebuttal has fully addressed your concerns, we would be grateful if you might consider increasing your score to reflect a solid accept.
>
> Best regards,
>
> Authors

---

### Note · Authors · 2025-08-13

Once again, we would like to thank all reviewers for their time and thoughtful feedback. In summary, in response to all of the helpful comments, we have

- Conducted ablation studies on path-weight sharing and CP decomposition;
- Evaluated CP decomposition across multiple ranks;
- Applied CP decomposition to an alternative architecture (NequIP);
- Analyzed the time complexity of the backward pass and measured backward pass runtimes of CP decomposition.

We are pleased to hear that we were able to address the reviewers' questions adequately.

---

### Decision · Program_Chairs · 2025-09-17

**Decision:**

Accept (poster)

**Comment:**

This paper proposes tensor decomposition networks (TDNs) that replace Clebsch-Gordan tensor products in SO(3)-equivariant neural networks with low-rank CP decompositions and a path-weight sharing strategy. The contribution is motivated by the high computational cost of equivariant tensor products in machine learning force fields, and the authors provide both theoretical guarantees (error bounds and universality) and strong empirical validation. Reviewers agree that the problem is important, the technical development is sound, and the experiments convincingly demonstrate significant efficiency gains with little to no accuracy loss.

Concerns were raised regarding the manual choice of CP rank, the approximate nature of equivariance, and the scope beyond Clebsch-Gordan tensor products. The authors responded with additional ablation studies, backward pass analysis, CPU runtime benchmarks, and a broader discussion of possible extensions. While some reviewers maintained borderline scores, they acknowledged that their questions were satisfactorily addressed. One reviewer raised their rating to clear accept, emphasizing the significance of the work for both AI and computational material science communities.

Overall, this is a technically solid paper with a well-motivated contribution and substantial potential impact. The methodological limitations are acknowledged and do not diminish the paper’s core value.